# Ex vivo engineered human plasma cells exhibit robust protein secretion and long-term engraftment in vivo

Rene Yu-Hong Cheng[1,2], King L. Hung [1,3], Tingting Zhang[1], Claire M. Stoffers[1], Andee R. Ott[1], Emmaline R. Suchland[1], Nathan D. Camp [1], Iram F. Khan[1], Swati Singh[1], Ying-Jen Yang [4], David J. Rawlings [1,5,6] ✉ & Richard G. James [1,2,5,7,8] ✉

Due to their unique longevity and capacity to secrete high levels of protein, plasma B cells have the potential to be used as a cell therapy for protein replacement. Here, we show that ex vivo engineered human plasma cells exhibit single-cell RNA profiles, scanning electron micrograph ultrastructural features, and in vivo homing capacity of long-lived plasma cells. After transferring human plasma cells to immunodeficient mice in the presence of the human cytokines BAFF and IL-6, we observe increases in retention of plasma cells in the bone marrow, with engraftment exceeding a year. The most profound in vivo effects of human IL-6 are observed within 20 days of transfer and could be explained by decreased apoptosis in newly differentiated plasma cells. Collectively, these results show that ex vivo engineered and differentiated human plasma cells have the potential for long-lived in vivo protein secretion, which can be modeled in small animals.

Biologic protein drugs are in use for the treatment of most classes of human diseases, including cancer, autoimmunity, and protein deficiencies[1]. A major challenge in the development of non-antibody biologics is poor pharmacokinetics and a short half-life. Conjugation of biologics with carrier devices such as PEG[2], Fc domains[3], albumin[4], or nanoparticles[5] can increase half-life, but these drugs still require recurrent delivery. An alternative method to conjugation is to express and deliver protein drugs using a long-lived cell source. One possible source is human long-lived plasma cells (PCs), a terminally differentiated B cell population that resides in the bone marrow (BM) for decades[6,7], and stably produces high concentrations of antibodies[8]. There are efforts pursuing several strategies to re-purpose PCs as a drug-secreting cell therapy for the treatment of disease[9–16]. The Voss[16] and Barzel[14] laboratories delivered engineered memory-like antigen-specific B cells to mice and used immunization to direct in vivo

differentiation to PCs[16]. After three rounds of immunization, antigen-specific PCs were quantifiable in BM using sequencing up to 250 days following B cell delivery. Alternatively, the Taylor lab[15] showed that in vitro differentiated engineered antigen-specific PCs were detectable in vivo up to 87 days following cell transfer.

Despite prior demonstrations of engineered PC engraftment in vivo, a major challenge that remains is engineering long-lived engraftment. Despite the longevity of a subset of PCs in prior studies with either in vitro[15] or in vivo[14,16] maturation of engineered PCs, engineered antibody titers decreased rapidly following PC elicitation. These data imply that as a field, we still lack a complete understanding of the factors needed to elicit an engineered population of long-lived PCs that are comparable in capacity and longevity to a vaccine-elicited long-lived PC[17,18]. A small minority of plasmablasts (PBs)/PCs that traffic to the BM are thought to become established long-lived PCs[7].

[1]Center of immunotherapy and Immunity, Seattle Children Research Institute, Seattle, WA 98101, USA. [2]Molecular Engineering and Science Institute, University of Washington, Seattle, WA 98195, USA. [3]Center for Personal Dynamic Regulomes, Stanford University School of Medicine, Stanford, CA 94305, USA. [4]Department of Applied Mathematics, University of Washington, Seattle, WA 98195, USA. [5]Departments of Pediatrics, University of Washington, Seattle, WA 98195, USA. [6]Department of Immunology, University of Washington, Seattle, WA 98195, USA. [7]Department of pharmacology, University of Washington, Seattle, WA 98195, USA. [8]Brotman-Baty Institute for Precision Medicine, Seattle, WA 98195, USA. ✉e-mail: drawling@uw.edu; rickerj@uw.edu

Consequently, long-lived PCs may not be intrinsically long-lived, but instead, require factors delivered locally in the BM to support longevity. Multiple types of PC survival factors are present in the BM, including extracellular matrix and cytokines or chemokines produced by stromal cells, dendritic cells, megakaryocytes, and eosinophils. One key pro-survival factor that supports PC longevity is Interleukin 6 (IL-6). However, mouse IL-6 does not efficiently activate the human IL-6 receptor[19,20], a feature that has impeded the modeling of human PC function in vivo in mouse systems.

We engineered human B cells to express non-antibody therapeutics, differentiated them into PCs and engrafted them for weeks into immunodeficient mice[9]. However, similar to other engineered PC experiments, the engrafted PCs exhibited limited longevity in vivo. Although ex vivo differentiation of engineered PCs offers significant advantages for the delivery of non-antibody therapeutics, including precise dose-control, quiescence in the cell product, and low risk of mutation in the therapeutic protein sequence, the method will only be successful with the development of a process that can elicit molecular features that lead to differentiation into long-lived, BM-resident PCs.

Here, we use single-cell RNA sequencing, electron microscopy, and in vivo modeling to show that long-lived engraftable PCs represent the ~30% of ex vivo-derived cultures that are CD138$^+$ and CD38$^{hi}$. We found that this sub-population of ex vivo generated PCs exhibit structural and transcriptional features of PCs isolated from human BM. These PCs are capable of survival for greater than 1 year in mice engineered to express hIL-6. Finally, we enumerate long-lived human PCs in mice and estimate that ~50,000 PCs are sufficient for the production of therapeutically relevant antibody titers (10 μg/mL).

## Results

### Single-cell transcriptomics illustrates the heterogeneity of ex vivo differentiated human PCs

Previously, we found that the differentiation of human B cells into cultures containing antibody-secreting cells results in a heterogeneous mixture of phenotypes[9]. To better understand this heterogeneity in ex vivo differentiated PCs, we evaluated B cells isolated from peripheral blood mononuclear cells (PBMC) from two donors immediately following cell expansion with T cell cytokines (day 7, Fig. 1a) and following differentiation (day 13, Fig. 1a) in culture using cellular indexing of transcriptomes and epitopes (CITE-seq; Fig. 1b, Supplementary Fig. 1a, and Supplementary Table 1). To describe the differentiation of human B cells, we used PAGA for trajectory inference and the Louvain-graph-based algorithm to cluster cells, including both gene expression and CITE-seq surface marker quantification. We chose the root cluster to be that with the highest expression of the B cell markers MS4A1, CD19, IGHD, MHC-class II, and lowest expression of the PC markers CD38 and IgG[21,22]. We found that cells from each donor were detected at similar proportions in all clusters, which suggested that donor-donor variability did not contribute significantly to the overall results. Additionally, while cells from day 7 and day 13 were generally grouped into time-specific subclusters (Fig. 1b, i and Supplementary Fig. 1a), we found some cells from the day 13 samples in the clusters predominantly containing day 7 cells and vice versa, which indicated that differentiation into PCs is ongoing. The clusters containing day 7 cells expressed markers of activated B cells (cluster 1 and cluster 6), B-blast (proliferating pre-PB, cluster 2), and pre-PB (cluster 4). In contrast, we found day 13 cells within interferon-stimulated activated B cells (cluster 3) and PB/PCs (cluster 0 and cluster 5). To verify the identity of the clusters, we plotted B cell programming master transcription factors (TFs): PAX5 (B cell TFs), EZH2 (Pre-PB TF), IRF4, and XBP1 (PC differentiation TFs) (Fig. 1c).

We observed two primary trajectories (Fig. 1b, ii). The first trajectory closely followed canonical B cell differentiation, starting from cells with markers of B cell activation (Fig. 1d, AICDA in Fig. 1c), MHC-class II (Fig. 1d), and proliferation (MKI67, Fig. 1c); this trajectory

includes PB and PC clusters with high expression of various immunoglobulins (Fig. 1d). As expected, surface marker expression via CITE-seq shows that B cell receptor (BCR/ anti-IgM oligos) expression decreased along pseudotime while the PC marker, CD38 (anti-CD38 oligos) increased (Fig. 1e). The major trajectory of B cells from our culture system aligns closely with PC development. In the second and more minor trajectory (cluster 1/3), day 13 cells retain an activated B cell-like signature, however, the predominant differences between day 7 and day 13 are interferon-induced genes, presumably due to IFN-alpha included in the B cell differentiation media (Fig. 1d).

Ex vivo differentiated PC cultures contain a heterogeneous mixture of phenotypes, including IgM$^{hi}$CD38$^{lo}$, IgM$^{lo}$CD38$^{lo}$, CD38$^{hi}$CD138$^{lo}$, and CD38$^{hi}$CD138$^{hi}$. Using a CITE-seq panel detecting these surface markers, we superimposed the phenotype subsets onto the gene expression UMAP (Fig. 1f, g). As expected, we found the majority of IgM$^{hi}$CD38$^{lo}$ cells mapped to clusters expressing an activated B cell signature. A small subset of these cells mapped to IgM PCs (clusters 3, 5), which indicates that these antibody-secreting cells (ASC) likely retain surface BCR. We found that the UMAP clusters were driven predominantly by unique immunoglobulin isotypes and light chains (Supplementary Fig. 1b, c) and the CD38$^{hi}$CD138$^{lo}$ and CD38$^{hi}$CD138$^{hi}$ populations were scattered within those subclusters. One exception was that IgA and IgM PCs exhibited proportionally fewer CD38$^{hi}$CD138$^{hi}$ cells (Fig. 1f, g).

We next used gene set enrichment analysis to compare CD38$^{hi}$ to CD38$^{lo}$ cells within ex vivo differentiated PCs. We found that CD38$^{hi}$ are enriched for transcriptional signatures found in PC datasets[8,23,24]. Examples of genes enriched in CD38$^{hi}$ (Fig. 1h) are those that regulate secretory organelles (endoplasmic reticulum, golgi, and secretory granule membrane) and antibody production (immunoglobulin, protein transport) (Supplementary Fig. 1b–d). As expected, canonical PC genes, including CD38, CD138 (SDC1), SLAMF7, BCMA, PRDM1, and XBP1 (Fig. 1h). We next compared CD38$^{hi}$CD138$^{lo}$ and CD38$^{hi}$CD138$^{hi}$ cells and found that SDC1 (CD138), several IgG isotypes, and several adhesion molecules are enriched in CD138hi cells (Fig. 1h and Supplementary Fig. 1e). Collectively, these transcriptional features of ex vivo cultured human B cells align with our prior understanding of B cell differentiation into PCs[8,23,24].

### A subset of ex vivo differentiated PCs resemble bone marrow ASCs

Despite literature showing that mitochondrial metabolism is important for PC differentiation[25–27], the single-cell sequencing analysis did not unearth transcriptional changes associated with mitochondria. To address whether this signature PC feature is present in ex vivo differentiated PCs, we measured mitochondrial volume and stress in differentiated B cell subsets using the vital dyes MitoTracker Green and MitoTracker Red CMXRos, respectively (gating strategy, Fig. 2a, b and Supplementary Fig. 2a, b). MitoTracker Green staining was greater in CD138$^{hi}$ cells relative to CD38$^+$ cells. On the other hand, MitoTracker Red CMXRos staining was higher in differentiated B cells (CD38$^{++}$ and CD138$^{hi}$) compared to active B cells/blast (CD38$^-$ and CD38$^+$), which indicates that mitochondrial membrane potential increases as a function of ex vivo differentiation of B cells into PCs.

We next investigated whether ex vivo differentiation of B cells led to morphologies exhibited by human PCs. Transmission electron microscopy (TEM) showed that ~50% of day 13 differentiated B cells have increased endoplasmic reticulum content relative to day 2 activated B cells (representative image, Fig. 2c; quantification, Fig. 2d). We also observed condensation of mitochondria in the ex vivo differentiated PCs (Fig. 2c and Supplementary Fig. 2c), which is indicative of active respiration[28]. These observations are consistent with previous observations of PC morphology[29] and demonstrate that ex vivo differentiated PCs share morphological features with human BM PCs.

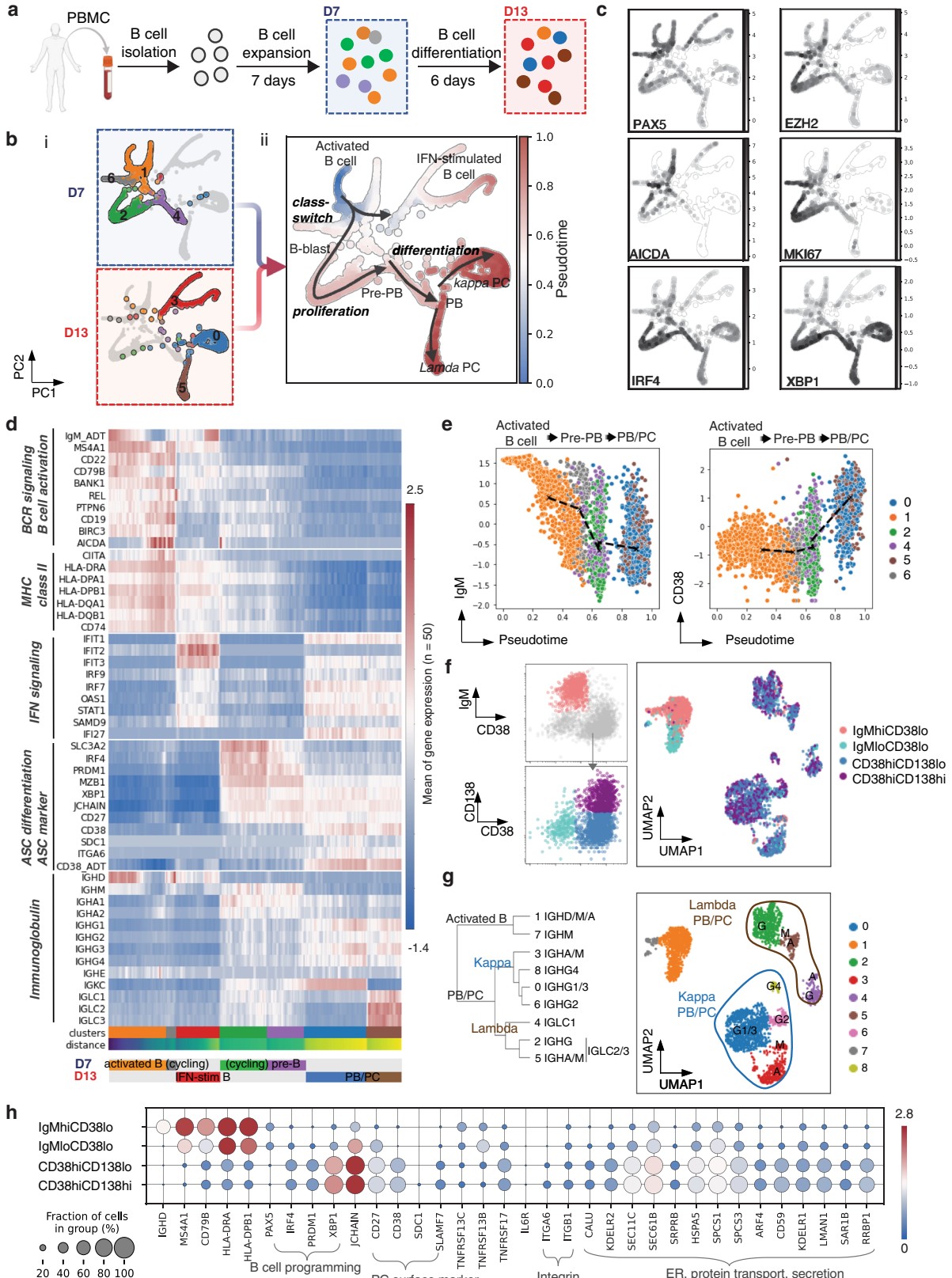

## Ex vivo differentiated PCs home to the BM and CD138 + cells stably engrafted in an immunodeficient mouse model

One of the important long-lived PCs features is homing to the BM[8,30]. To examine if ex vivo differentiated PCs can home to and be retained in the BM, we designed a gene-editing-based luciferase reporter that could be monitored in vivo. We introduced the firefly luciferase coding sequence into an AAV-based repair template with homology arms that facilitate insertion and expression within the CCR5 locus following CRISPR-mediated gene editing[9] (Fig. 3a). We isolated and activated human B cells and introduced firefly luciferase using this template for homology-directed repair (HDR)-based gene addition. After differentiation of edited cells into PBs and PCs, we isolated genomic DNA

**Fig. 1 | Single-cell transcriptomics illustrate the heterogeneity of ex vivo differentiated human PCs. a** Schematic of B cell culture ex vivo, and experimental setup: day 7 and day 13 cells were collected for CITE-seq. It was partially created with BioRender.com. **b** Single-cell graph by PAGA trajectory analysis of day 7 and day 13 cells, and heatmap by inferred pseudotime of all cells ($n = 6011$) from both time point and two biological replicates. Trajectory starts from the initial state node (cluster 1, activated B cells from day 7 cells) and ends at terminal state nodes (cluster 0/5, kappa/lambda PCs from day 13 cells). Cell differentiation states in between are labeled according to their representative cell status. **c** Single-cell trajectory graph heatmap showing expression of representative genes which indicate each cell status. **d** Expression heatmap of Louvain clusters, gene sets in each pathway show gene regulation along B cell development. Each data point present in

the heatmap is computed by the average of normalized gene expression from 50 cells. **e** Surface protein (antibody-derived tag) expression trend along pseudotime. BCR expression is downregulated and CD38 expression is upregulated during PC differentiation. **f** Classification of B cell subsets categorized by the indicated protein markers: IgM^hiCD38^lo, IgM^loCD38^lo, CD38^hiCD138^lo, and CD38^hiCD138^hi. UMAP dimension-reduction projection of day 13 B cells ($n = 2897$) from two biological replicates. **g** Predominant immunoglobulin expression in each subset from **f**. **h** Dotplot visualization of day 13 B cells ($n = 2897$): subsets are listed on y-axis and genes (features) are listed along the x-axis. Dot size represents the percentage of cells in a group expressing each gene; the dot color indicates the mean expression level in a group.

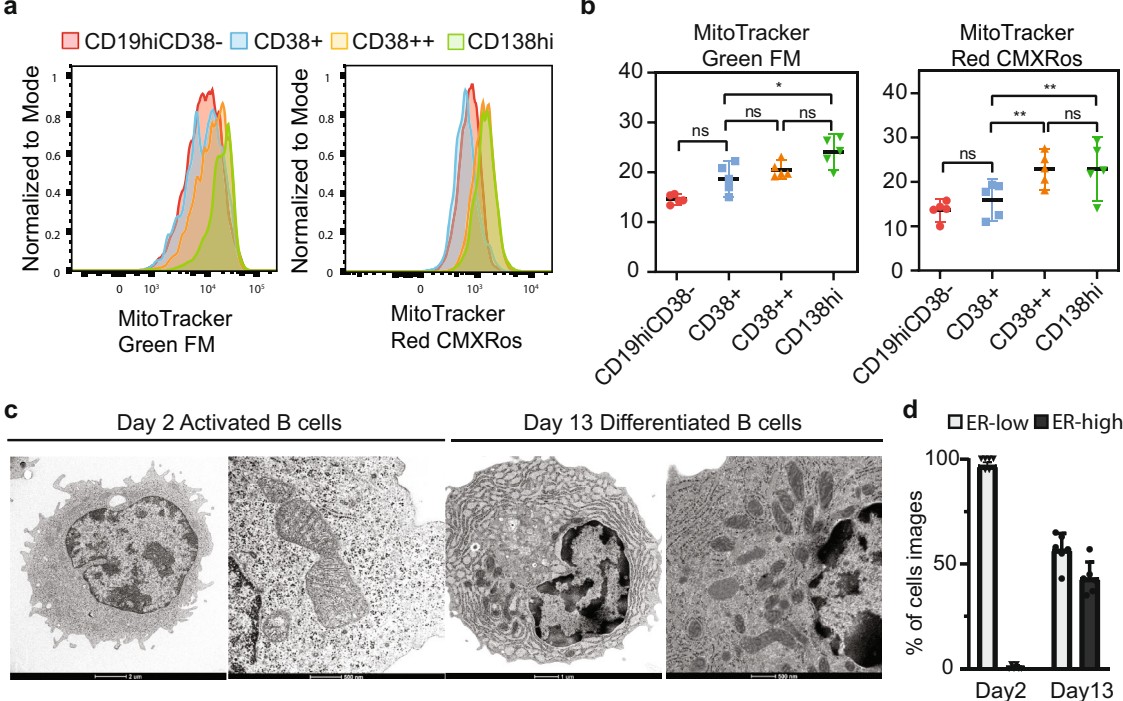

**Fig. 2 | A subset of ex vivo differentiated PCs resemble bone marrow ASCs. a, b** Using the indicated protein markers: active B (CD19^hiCD38^−), CD38^+, CD38^++, and CD138^hi flow cytometry showing expression of MitoTracker Green FM (mitochondrial volume) and MitoTracker Red CMXRos (mitochondrial stress) from different B cell subsets. The data ($n = 5$ subjects) are normalized by minimal and maximal intensity per replicate and presented as mean and 95% confidence intervals. To assess significance, we used a paired one-way ANOVA with Tukey's multiple comparison test (*$p < 0.05$, **$p < 0.01$, ***$p < 0.001$). P values for CD38^+ and CD138^hi

with MitoTracker Green FM is 0.0123, for CD38^+ and CD38^++, CD38^+ and CD138^hi with MitoTracker Red CMXRos are 0.0044 and 0.0041, respectively. Source data are provided as a Source Data file. **c** Transmission electron microscopy of day 2 and day 13 B cells from three-stage culture. **d** Quantification of rough endoplasmic reticulum (RER) in ex vivo differentiated cultures. Day 2 (43 images) and day 13 cells (60 images) were scored blindly for having prominent RER or not ($n = 6$ participants). Data were presented as mean ± SD. Source data are provided as a Source Data file.

from a subset of cells and quantified HDR using ddPCR; 18.63% of alleles exhibited integration of the luciferase repair template.

Luciferase-engineered PCs were transferred into NOD.Cg-*Prkdc^scidIL2rg^tm1Wjl*/SzJ (NSG) immune-deficient mice (10 million cells per recipient), and subsequently tracked using in vivo imaging (Fig. 3a schematic). Two days post-transfer, gene-edited cells accumulated in the hindlimbs and chest region (Fig. 3b, c and Supplementary Fig. 3a). The signal localized to the chest diminished over time, likely reflecting the initial trafficking of transferred cells to the lung or liver. In contrast, the signal in the hindlimbs and the sternum remained stable from day 15 to day 51 (Fig. 3b, c and Supplementary Fig. 3a). Furthermore, human immunoglobulin G (hIgG) and human immunoglobulin M (hIgM) dynamics are consistent with luciferase signal (Fig. 3d and Supplementary Fig. 3b, c). To measure the functional capabilities of the transferred PCs, we quantified hIgG by ELISA for up to 1 year after transfer. Consistent with the luciferase imaging, hIgG peaked 2 weeks post-transfer, diminished rapidly over the subsequent month, and

subsequently stabilized (Supplementary Fig. 3c) at concentrations between 1000–3000 ng/ml for a year post-adoptive cell transfer (Supplementary Fig. 3c). These results demonstrate that ex vivo-derived B cell populations can home to the BM, and that a subset of these cells reside within the BM and stably produce antibodies for as long as 1 year.

To investigate which populations in the ex vivo cultures are responsible for long-term in vivo engraftment, following ex vivo differentiation, we enriched for CD138^+ and CD138^− cells and infused these subpopulations into NSG mice (Fig. 3e). We found that the CD138^+ cells engrafted similarly to unenriched cells, and that CD138^− cells engrafted poorly (Fig. 3f, g). To investigate the differences between CD138^+ and CD138^− PCs, we used bulk RNA sequencing. As we observed in the CITE-seq data (Supplementary Fig. 3d), immediately after differentiation, several apoptotic genes were downregulated in CD138^+ PCs (Supplementary Fig. 3e). Furthermore, upon the culture of the CD138^+ cells for 6 additional days, we found that the PC homing

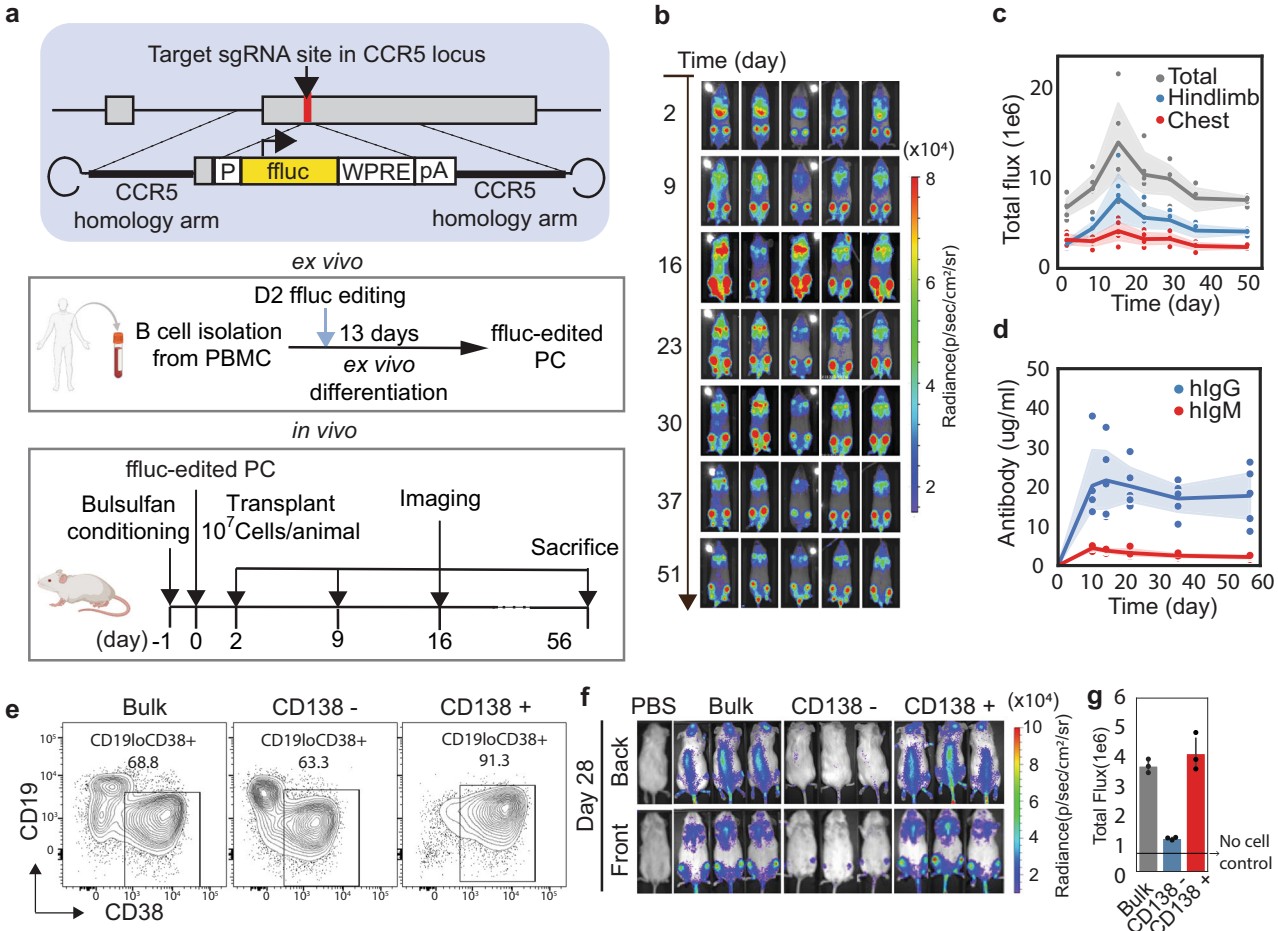

**Fig. 3 | Ex vivo differentiated PCs home to BM and CD138+ PC is the main population engrafted. a** Schematic of firefly luciferase showing the homology-directed repair strategy, the B cell engineering and differentiation strategy, and the in vivo engraftment and tracking strategy. It was partially created with BioRender.com. **b** Representative images from an in vivo engraftment experiment tracking engineered B cells from Day 0 to day 51 using firefly luciferase imaging by IVIS (n = 10 animals, two independent donors. The imaging data is representative from a single donor; the data from donor two is in Supplementary Fig. 3a, b).

**c** Luciferase flux was quantified in the hindlimb and chest at each of the time points. **d** Immunoglobulin was quantified in serum by ELISA. **c, d** The line indicates the mean at each time point, and the shadow shows the 95% confidence intervals. Source data are provided as a Source Data file. **e** Flow cytometry of cell subsets after enrichment with CD138 antibody. **f** Representative images and **g** quantification from in vivo subset engraftment tracking engineered B cells after a month. Cells were edited to express firefly luciferase and were imaged by IVIS (n = 3 animals). Data were presented as mean ± SD. Source data are provided as a Source Data file.

markers CXCR4, CXCR6, and LFA-1[8,30], are increased in CD138 + PCs (Supplementary Fig. 3f). Finally, negative (SOCS3; ref. [31]) and positive (IL6R) regulators of IL-6 signaling, were decreased and increased respectively (Supplementary Fig. 3f), suggesting that IL-6 signaling was upregulated in cultured PCs. Collectively, transcriptional changes in CD138+ PCs likely contribute to the engraftment and survival of ex vivo differentiated PCs in vivo.

## Human IL-6 and BAFF promote the engraftment of ex vivo differentiated PCs

We hypothesized that the decline in hIgG production over time in the mouse model might be partly explained by the lack of key human cytokines produced by the BM of NSG mice. For example, mouse IL-6, a cytokine critical for PC longevity[6], does not efficiently activate human IL6R[19,32]. To test whether hIL-6 can improve survival or engraftment of ex vivo differentiated human PCs in NSG mice, we created a humanized IL-6 mouse model by delivering hIL-6 to neonates via lentiviral injection (schematic, Fig. 4a). Again, we generated ex vivo differentiated PBs and PCs, and adoptively transferred these cells into either control or similarly aged NSG recipients that were engineered via neonatal transduction with lentivirus expressing hIL-6 (serum hIL-6 quantified in Supplementary Fig. 4a). hIgG production by engrafted PBs and PCs

in mice expressing hIL-6 was significantly increased ~3-fold at all time points (Fig. 4b and Supplementary Fig. 4b). Furthermore, the antibody levels are highly correlated with hIL-6 in mice serum (Pearson correlation in Supplementary Fig. 4c, quantified antibodies in Fig. 4b and Supplementary Fig. 4e, hIL-6 in Fig. 4d).

human BAFF (hBAFF) is a secreted protein produced by several myeloid subsets and B cells that enhances hIgG production by PCs in vivo[7,33,34]. To mimic local hBAFF production, we used HDR-based editing to introduce hBAFF and a cis-linked GFP reporter into human B cell cultures via delivery to the CCR5 locus as previously described[9]. B cells edited to express hBAFF exhibited several significant changes in culture, including higher proportions of CD38 + cells, IgG+ cells, and increased proliferation relative to B cells edited to unedited B cells (Supplementary Fig. 5). We found that hBAFF-edited B cells transferred into NSG mice stably secreted higher levels of hIgG than unedited B cells (Fig. 4b, c). Additionally, B cells expressing hBAFF transferred into hIL-6-expressing mice produced higher quantities of hIgG (Fig. 4c), hIgM (Supplementary Fig. 4e), and hBAFF (Fig. 4d) than those introduced into control NSG mice, indicating that hIL-6 and hBAFF acted in an additive way to promote PC longevity and/or protein production. Together, these findings demonstrate that provisioning NSG mice with human cytokines enables long-term engraftment (>1

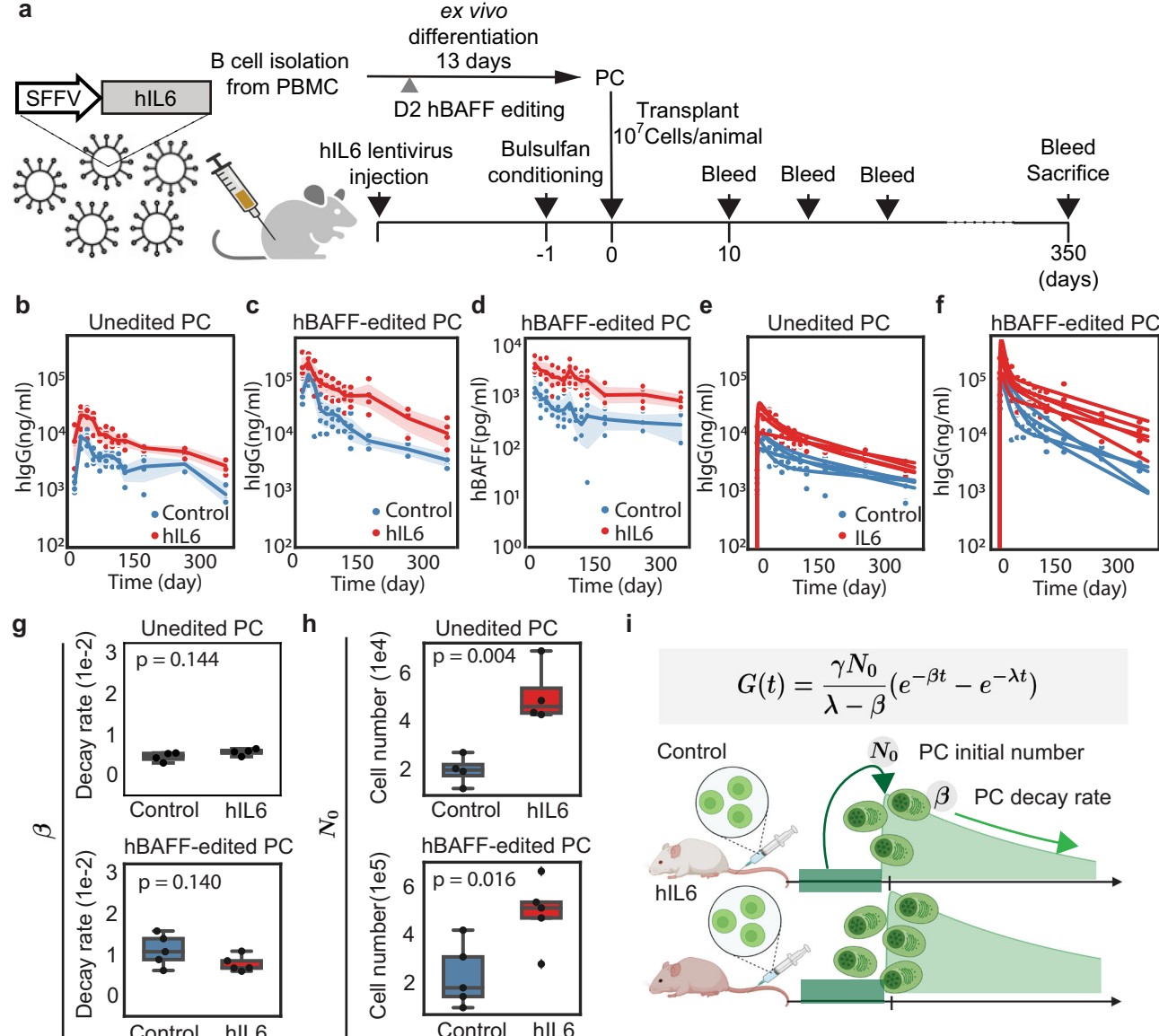

**Fig. 4 | hIL-6 and hBAFF promote engraftment of ex vivo differentiated PCs.**
**a** Schematic describing in vivo infusion of ex vivo differentiated B cells. Mock
($n = 4$ animals) or hBAFF ($n = 5$ animals) edited cells were transferred into NSG
mice engineered to express hIL-6 (hIL-6·NSG). It was partially created with
BioRender.com. **b**–**d** ELISA was used to quantify B cell production of hIgG and
hBAFF in NSG or NSG·hIL-6 animals. The line indicates the mean at each time
point, and the shadow shows the 95% confidence intervals. Source data are
provided as a Source Data file. **e**, **f** Using antibody secretion rate and antibody
degradation rate as parameters (Supplementary Table 2), we fitted curves to
explain the observed hIgG dynamics in the indicated mouse models. **g**, **h** For
each animal (Mock, $n = 4$ animals; hBAFF, $n = 5$ animals), we used the fitted curves
to calculate the decay rate and engrafted PC numbers. The box plot represents a
central line denoting the median value (50th percentile), while the box contains
the 25th to 75th percentiles of the dataset with black whiskers marking the
maximum (95th percentile) and minimum (5th percentile). *P* values were cal-
culated using an unpaired two-tailed Welch's *t*-test. **i** Schematic showing how the
key parameters (initial engraftment number and decay rate) are altered by hIL-6.
It was created with BioRender.com.

year) and antibody production (>10 μg/mL) by ex vivo differentiated
human PCs.

Following an adoptive transfer of ex vivo differentiated human
PCs, hIgG abundance rose sharply in the first 14 days post-engraftment.
hIgG levels then decreased rapidly and stabilized starting at ~60 days
post-engraftment. We hypothesized that distinct PC cell populations
are responsible for this complex dynamic: a relatively large number of
short-lived PCs contribute to the early increases in hIgG, whereas a
smaller number of long-lived PCs are responsible for stable hIgG pro-
duction. We created mathematical models to explain each of these
populations. To do so, we denoted the number of PC as $N$ and the

amount of antibody hIgG as $G$. Their dynamics can be modeled by the
following equations.

$$\frac{dN}{dt} = -\beta N \tag{1}$$

$$\frac{dG}{dt} = \gamma N - \lambda G = \gamma N_0 e^{-\beta t} - \lambda G \tag{2}$$

where $\beta$ represents the PC decay rate, $\gamma$ represents the hIgG secretion
rate, and $\lambda$ represents the clearance rate of hIgG. In the solution of Eq.2,

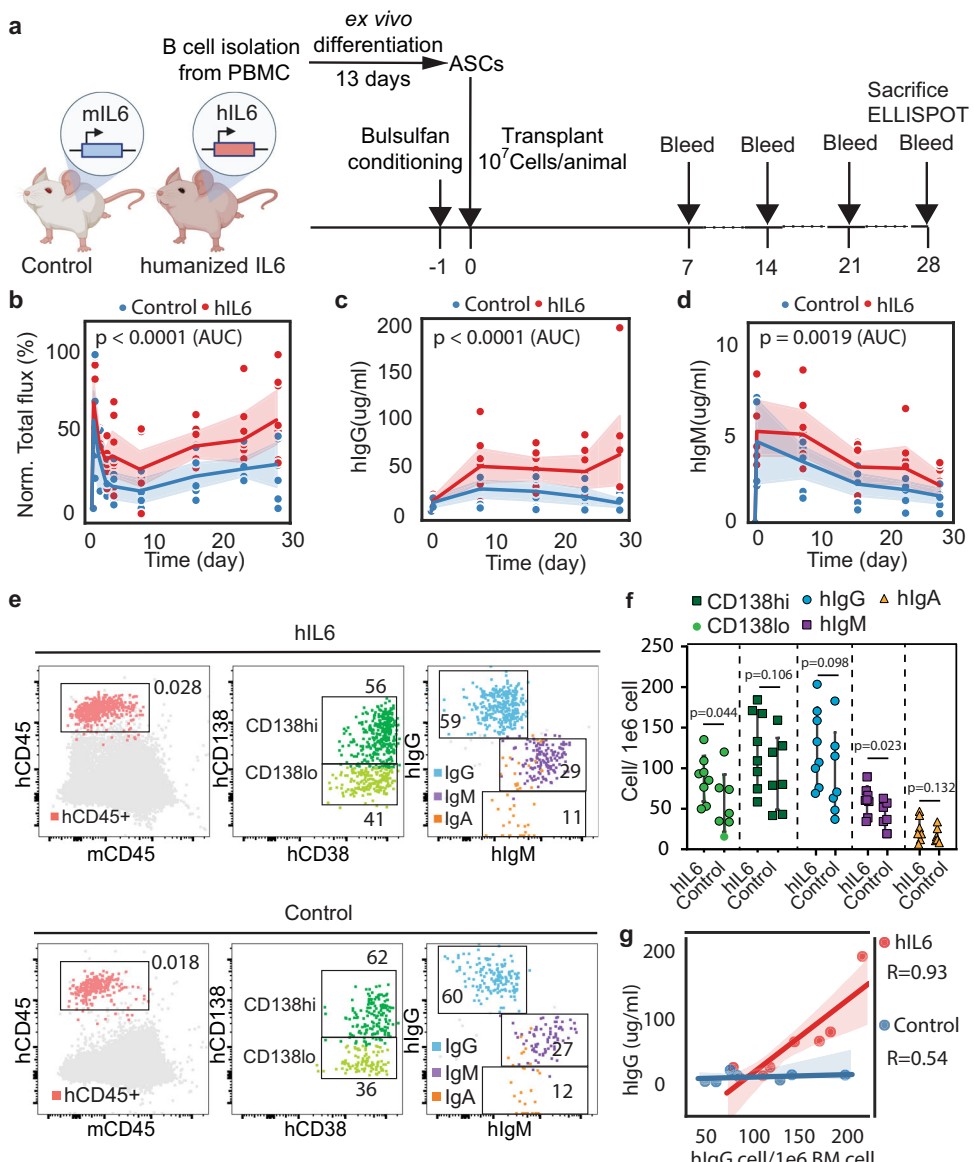

**Fig. 5 | Transfer of PCs into an immunodeficient mouse model expressing hIL-6 improves PCs engraftment. a** Schematic describing the transfer of ex vivo-derived PCs into B-NDG (control; $n = 7$ animals from two independent experiments) and hIL-6-B-NDG knock-in (hIL-6; $n = 8$ animals from two independent experiments) mice. It was partially created with BioRender.com. **b–d** Firefly luciferase was quantified by IVIS imaging (normalized by Min-max normalization), hIgG and hIgM were quantified by ELISA and each point was plotted individually. The shadow represents the 95% confidence interval and the line represents the mean. *P* values were calculated in an unpaired two-sided *t*-test by comparing the area under curve (AUC) from individual mice. *P* values for normalized flux AUC and IgG AUC are 1.46E-05 and 1.37E-05, respectively. Source data are provided as a Source Data file. **e, f** Representative (**e**) and quantified (**f**) flow cytometry of BM cells from sacrificed mice using the indicated antibodies (control; $n = 7$ animals from two independent experiments) and hIL-6-B-NDG knock-in (hIL-6; $n = 8$ animals from two independent experiments). Data were presented as mean ± SD. *P* values were calculated using an unpaired one-sided *t*-test. Source data are provided as a Source Data file. **g** The relationship between hIgG antibody concentration and hIgG PCs counts in B-NDG mice and hIL-6-B-NDG mice. The correlation between them is presented by the Pearson correlation coefficient based on their quasi-linear relationship and assumption of normality of variables. The line indicates the mean at each time point, and the shadow shows the 95% confidence intervals. Source data are provided as a Source Data file.

the amount of antibody at a given time ($t$) is

$$G(t) = \frac{\gamma N_0}{\lambda - \beta}\left(e^{-\beta t} - e^{-\lambda t}\right) \qquad (3)$$

The parameters $\gamma$, $\lambda$ are described previously[35–38]. We used Eq. 3 to model the PC behavior and fit the initial number of PCs that survive transit through the lung/liver, and/or differentiate and survive in vivo beyond 21 days (denoted as $N_0$) and the decay rate ($\beta$) of these cells (Supplementary note, model 1, Supplementary Table 2, and Supplementary Fig. 6a). In this model, we explicitly ignore short-lived PCs that

produce antibodies initially following transfer, but are not stably retained. We applied this model to discriminate between the experimental conditions (Fig. 4e, f). Based on the fitted results, the decay rate of long-lived PCs was similar in cells engrafted into hIL-6 expressing mice and those engrafted into control mice (Fig. 4g). However, hIL-6 increased the number of PCs capable of long-term engraftment that were retained in the BM immediately post-transfer (Fig. 4h and schematic, 4i). In contrast, cell-intrinsic expression of hBAFF appears to boost the initial PC numbers but this is accompanied by more rapid decay (Supplementary Fig. 6b). These data support prior reports showing that BAFF promotes the expansion of PBs[7,39,40]. Finally, based

on our estimates, ~500,000 PCs (~5% of the inoculum) are reproducibly retained in hIL-6 expressing mice, of which ~50,000 long-lived PCs are still present after 1 year (Supplementary Fig. 6c, d).

To investigate whether the mathematical model was supported by experimental data, we next investigated whether recipient NSG mice that express hIL-6 exhibited an increase in the proportion of PCs initially retained following engraftment. Again, we engineered ex vivo differentiated B cells to express luciferase and transferred these populations into hIL-6-transduced or control mice. Three days after infusion, we observed a significant increase in both the luciferase signal (Supplementary Fig. 6e, f) and the level of secreted hIgG (Supplementary Fig. 6g) in mice expressing hIL-6. Together, these observations support the conclusion that systemic hIL-6 primarily facilitates an increase in the number of PCs initially retained in the BM following adoptive transfer.

### Physiologic expression of hIL-6 promotes retention of BM-engrafted PCs

To explore features that regulate engraftment of ex vivo differentiated PCs in mice that express hIL-6 under physiological control, we made use of an immune-deficient model (NOD.CB17-$Prkdc^{scid}Il2rg^{tm1}$/Bcgen, B-NDG) with hIL-6 knocked into the mouse $Il6$ locus, NOD.CB17-$Prkdc^{scid}Il2rg^{tm1}Il6^{tm1(IL6)}$/Bcgen (humanized IL-6 B-NDG). Ex vivo differentiated PCs engineered to express luciferase were engrafted into hIL-6-expressing mice or control animals, and evaluated in several ways (schematic, Fig. 5a). As we observed in mice physiologically expressing hIL-6, luciferase detected from edited B cells was retained in the BM at higher rates in hIL-6 mice than in controls (Fig. 5b and Supplementary Fig. 7a). Similarly, we found that hIgG (Fig. 5c) and hIgM (Fig. 5d) levels were significantly higher in hIL-6 animals relative to control animals. In each experiment, we found that the differences in luciferase initially became apparent at 48 h following the transfer, suggesting that hIL-6 promotes the retention of engrafted PCs in the BM.

To evaluate the phenotypes of engrafted human B cells, we sacrificed the animals 28 days post-transfer, and used flow cytometry to characterize the human cells in the BM. As expected, the human cell engraftment (hCD45+) was higher in hIL-6 mice than in controls (Representative image Fig. 4e and quantified Supplementary Fig. 7b, c). Additionally, in both mouse hosts, most human cells exhibited features expected in PCs, including expression of CD38, CD138, and the antibody isotypes hIgA, hIgG, or hIgM (Fig. 5e, f). We found that the number of durably engrafted hIgG PCs was ~10,000/$10^8$ BM cells. In humanized mice, the number of hIgG+ PCs was highly correlated with hIgG abundance (Pearson's r = 0.93) in hIL-6 mice (Fig. 5g). Collectively, the luminescence, ELISA, and cell phenotype data indicated that hIL-6 increased retention of PCs and longevity in immune-deficient mice.

### hIL-6 increases survival of ex vivo differentiated PCs by inhibiting apoptosis of CD138$^{lo}$ PCs

We predicted that the boost in PC retention in the BM caused by hIL-6 was due to differences in death or proliferation of ex vivo differentiated PCs in animals upon transfer (Fig. 6a). To investigate how PC turnover is regulated by hIL-6, we differentiated PCs ex vivo using a three-stage culture system, isolated CD138+ PCs and cultured them for an additional 21 days in the presence or absence of hIL-6. We quantified the number of surviving PCs over time and found that cell numbers decreased in both conditions. However, hIL-6 significantly increased the number of viable PCs at each time point relative to vehicle (Fig. 6b). To uncover the dynamics of cell number decay, we first used $N(t)/N_0 = e^{-\beta t}$, where $N_0$ is an initial cell number, and $\beta$ is decay rate, to fit cell number over time. However, we found that $\beta$ as a parameter did not fit after the first 6 days of observation (Fig. 6c, dash line). We concluded that the decay rate is not constant during the 21-day culture. Therefore, to account for the changing decay rate, we used the

following equation (Supplementary note, model 2):

$$N(t)/N_0 = e^{-(\beta_0 t + \beta_1 t^2)} \qquad (4)$$

The resulting dynamics of fitted $\beta_0$ and $\beta_1$ matched the data (Fig. 6c, solid line), and we presented all the fitted dynamics of all samples with hIL-6 and without hIL-6 (Fig. 6b).

We found that PC cell numbers decreased most rapidly in the first six days post-differentiation and were stable thereafter (Fig. 6c and Supplementary note, model 2). Upon fitting the dynamics of PC numbers in these cultures, $\beta(t)$ (Fig. 6d and Supplementary Fig. 8), $\beta_0$ is the initial value of $\beta$. It can be interpreted as the acute effect of hIL-6, and later on shows the $\beta$ eventually converge to a similar level at day 21, which is shown the difference between the two $\beta$ slope ($\beta_1$). The converged $\beta$ matches the closeness of $\beta$ from the long-term model (Fig. 4g). Using this mathematical model, we found the nature of newly generated PCs are still undergoing further differentiation with a more anti-apoptotic ability or/and PCs without anti-apoptotic just died and selectively remained the anti-apoptotic PC. And that hIL-6 promoted early, but not later PC survival.

To examine whether hIL-6 affects the proliferation of ex vivo differentiated PCs or PBs, we repeated the three-stage differentiation culture, isolated CD138+ cells and CD138- cells, and further cultured each population for 72 h. At this point, we introduced BrdU and continued culturing for an additional 72 h. Neither differentiated PCs nor differentiated PBs exhibited substantial amounts of proliferation ex vivo (Fig. 6e; <1% of cells divided in this 72 h period), although we did find that BrdU labeling was significantly higher in CD138+ PCs in the presence of hIL-6.

Next, we directly measured the impact of hIL-6 on cell death. We quantified caspase activation using flow cytometry at several time points (1, 24, and 48 h) following the termination of the three-stage differentiation cultures. hIL-6 decreased active caspase levels in CD138$^{lo}$ PCs (CD38$^{hi}$CD138$^{lo}$; Fig. 6f), but not in active B cells or CD138$^{hi}$ PCs. We also found that relatively few CD138$^{hi}$ cells exhibited activated caspase, indicating that upon terminal differentiation to PCs, B cells exhibit increased survival. Together, these observations demonstrate that hIL-6 primarily regulates PC durability by increasing cell survival immediately following differentiation.

### hIL-6 increases per-cell secretion rate in PCs

In addition to the durability of PC engraftment, the per-cell secretion rate also contributes to total antibody production. To examine whether treatment with hIL-6 alters PC per-cell secretion rate, we differentiated PCs using the three-stage culture method, and cultured CD138+ cells for several time periods in media containing hIL-6 or vehicle. At each time point (0 days, 6 days, 11 days, 16 days, and 21 days; schematic Fig. 6g), we resuspended a fixed number of cells in fresh media, collected media for hIgG quantification by ELISA. We found that ex vivo differentiated PCs treated with hIL-6 exhibited significantly larger quantities of hIgG (Fig. 6h) per cell from day 13.5, with a mean value ~50 pg/cell/day. We next quantified protein production at a single cell level using hIgG ELISPOT from ex vivo differentiated PCs cultured for 21 days with or without hIL-6. For each spot, we calculated spot intensity and plotted the ELISPOT size distribution using cells derived from three donors (Fig. 6i). We observed substantial heterogeneity in protein production between ASCs (approximately tenfold; Supplementary Fig. 9). However, we found that PCs treated with hIL-6 had many more high-producing cells (Supplementary Fig. 9) and an increased mean spot size (Fig. 6j). To examine whether in vivo exposure to hIL-6 alters protein secretion on a per cell basis, we quantified

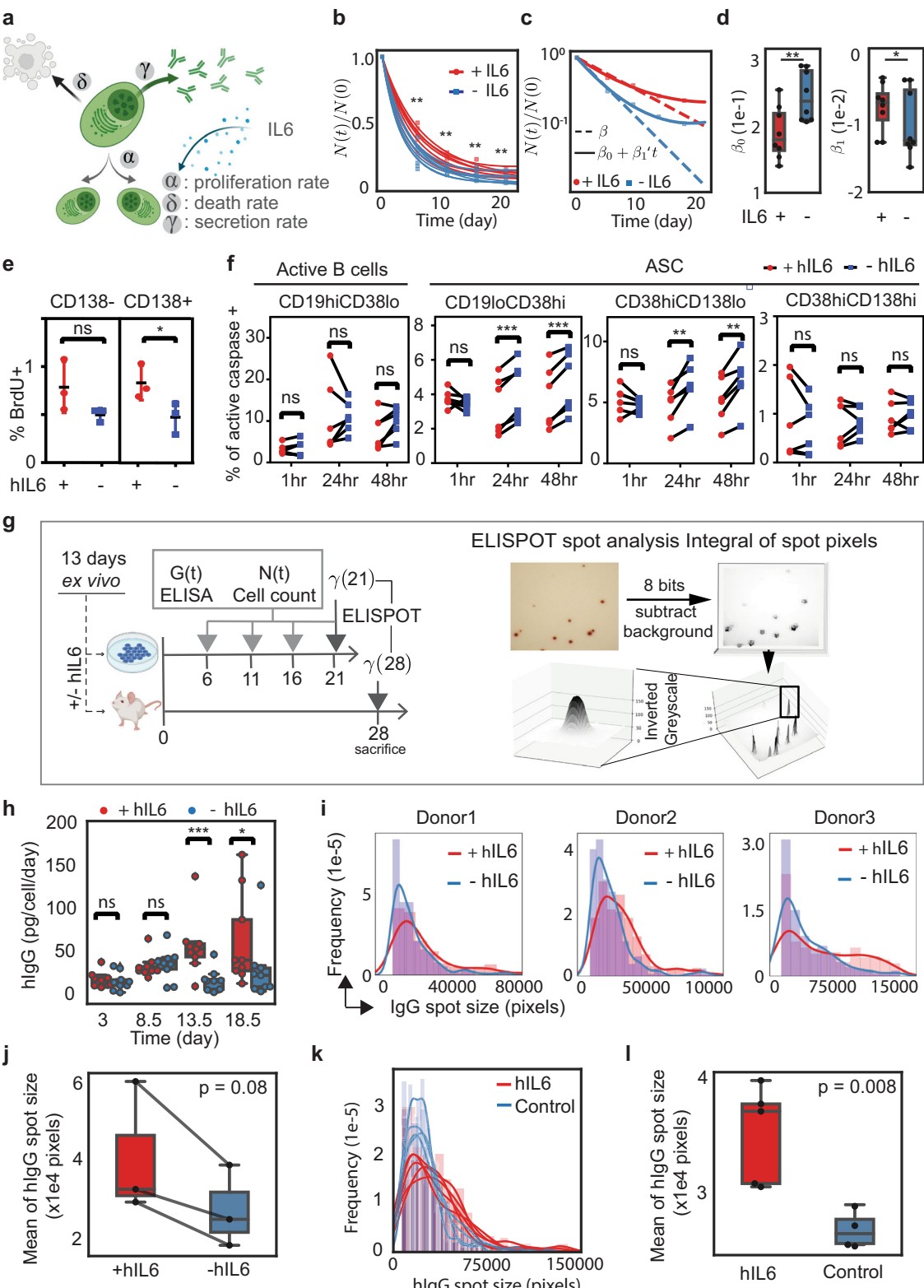

hIgG secretion from PCs isolated from the BM of mice engrafted with ex vivo differentiated PCs (see Fig. 4a). Similar to the ex vivo observations, PCs isolated from hIL-6 mice exhibited more high-producing cells (Fig. 6k) and increased mean spot size (Fig. 6l). Finally, we adjusted the assumptions of our mathematical model of PC engraftment to account for different antibody secretion rates in hIL-6 expressing mice (Supplementary Table 2). The remodeling results (Supplementary Fig. 10) did not alter the earlier conclusion that hIL-6

promotes the retention of PCs in BM. Collectively, these data demonstrate that exposure to hIL-6 increases the survival, retention in BM, and protein secretion of ex vivo differentiated PCs.

## Discussion

In this study, we show that human B cells can be differentiated ex vivo into PCs that exhibit multiple key features of long-lived PCs found within human BM, including the transcriptional and metabolic

**Fig. 6 | hIL-6 prevents CD38$^{hi}$CD138$^{lo}$ PC apoptosis and increases hIgG secretion rate. a** Schematic showing possible PC features affected by hIL-6, including proliferation rate ($\alpha$), death rate ($\delta$), and antibody secretion rate ($\gamma$). **b** Following ex vivo differentiation of PCs, CD138+ cells were isolated and cultured for 20 days in the presence or absence of hIL-6. The cell number for each replicate ($n = 8$; four independent donors) was plotted and curves were fitted (red and blue lines) based on the primary cell counting data. A two-sided paired $t$-test is performed for each time point, significance is under a significance level of $\alpha = 0.05$. (*$p < 0.05$, **$p < 0.01$, ***$p < 0.001$). $P$ values from left to right are 0.001, 0.007, 0.004, and 0.006, respectively. Source data are provided as a Source Data file. **c** Representative fitted curves with constant decay rate ($\beta$) and time-dependent decay rate ($\beta_0 + \beta_1/t$), respectively. **d** Fitted results of $\beta_0$ and $\beta_1$ from independent experiments ($n = 8$; four donors). The box plot represents a central line denoting the median value (50th percentile), while the box contains the 25th to 75th percentiles of the dataset with black whiskers marking the maximum (95th percentile) and minimum (5th percentile). $P$ values were calculated by a two-sided paired $t$-test. A two-sided paired $t$-test is performed, significance is under a significance level of $\alpha = 0.05$. (*$p < 0.05$, **$p < 0.01$, ***$p < 0.001$). $P$ value for $\beta_0$ and $\beta_1$ are 0.0027 and 0.03, respectively. **e** Ex vivo differentiated PCs were labeled with BrdU at day 16 for 72 h and analyzed by flow cytometry at day 19 ($n = 3$ donors). A two-sided paired $t$-test is performed, significance is under a significance level of $\alpha = 0.05$. (*$p < 0.05$, **$p < 0.01$, ***$p < 0.001$). Data were presented as mean ± SD. Source data are provided as a Source Data file. **f** Ex vivo differentiated PCs were incubated with or without hIL-6 for the indicated time periods and flow cytometry was used to quantify Caspglow in the indicated phenotypic subsets ($n = 6$ donors). $P$ values

were elucidated by a two-sided paired $t$-test (*$p < 0.05$; **$p < 0.01$, and **$p < 0.001$). Source data are provided as a Source Data file. **g** Schematic describing experimental outline for quantification of PC antibody secretion capacity ex vivo and in vivo. **h** hIgG secretion calculated from ELISA and cell counts (pg/cell/day) from day 0 to day 21 after three-stage culture ($n = 9$; five donors). $P$ values were calculated using a two-sided paired $t$-test, significance is under a significance level of $\alpha = 0.05$ (*$p < 0.05$; **$p < 0.01$, and **$p < 0.001$). $P$ values from left to right are 0.61, 0.70, 0.0007, and 0.015, respectively. Source data are provided as a Source Data file. **i** Histogram and kernel density estimate (solid line) distributions of ELISPOT size from ex vivo day 13 CD138+ cells cultured with or without hIL-6 for 21 days. Data were normalized by scaling input vectors individually to the unit norm. Source data are provided as a Source Data file. **j** Mean of ELISPOT spot in three-dimensional size from ex vivo differentiated PCs cultured for 21 days ($n = 3$ donors). $P$ values were calculated using a two-sided paired $t$-test. **k** Histogram and kernel density estimate (solid line) distributions of ELISPOT size converted from three-dimensional pixels, quantified from ASCs from B-NDG mice ($n = 4$ animals) and hIL-6-B-NDG mice ($n = 5$ animals) at day 28. Data normalized by scaling input vectors individually to unit norm. Source data are provided as a Source Data file. **l** Mean of ELISPOT spot in three-dimensional size from ASCs from B-NDG mice and hIL-6-B-NDG mice at day 28. $P$ values were calculated using a two-sided $t$-test. **h, j, l** Box plot represents a central line denoting median value (50th percentile), while the box contains the 25th to 75th percentiles of the dataset with black whiskers marking the maximum (95th percentile) and minimum (5th percentile). **a, g** were created with BioRender.com.

---

program, cell surface phenotype, and in vivo longevity and capacity for sustained endogenous protein production. The cell surface phenotype (CD19$^{lo}$CD38$^{hi}$CD138$^{+}$) and transcriptional features of ex vivo-derived PCs closely resemble human long-lived antigen-specific ASCs[6]. Most notably, we show that ex vivo differentiated PCs migrate to the BM, where they are retained and functional for >1 year, demonstrating longevity, as well as sustained exogenous protein production following adoptive transfer. Finally, we show that ex vivo differentiated human PCs are dependent on signaling from hIL-6 for retention in the BM, survival following differentiation, and optimal secretory capacity.

We demonstrate that ex vivo terminally differentiated PCs can survive in vivo for longer than a year in immunodeficient mice. Following vaccination, endogenous human PCs can compete into, and be retained within the BM niche well into adulthood[41]. Our data imply that immunodeficient mice lack key features that promote human PC survival in the BM niche, and thus underestimate the longevity of ex vivo-derived PCs. Supporting this idea, we found that the half-life of human PCs in immunodeficient mice is ~138 days (based on decay rate, 0.005), which is comparable to mouse PC longevity in response to immunization[17,18]. In larger animals like rhesus macaque[42] and human beings[43], tetanus-specific PCs have a half-life of ~1350 days and ~4000 days, respectively. Therefore, we hypothesize that transfer of autologous-engineered human PCs will likely exhibit higher longevity in human hosts than in immunodeficient mice.

A possible reason that human PCs exhibit decreased engraftment capacity in mice is imperfect interactions between mouse cytokines and human receptors. Proteins involved in PC homing, like CXCL12 and its receptor CXCR4, exhibit high degrees of homology (90 and 92% amino acid conservation, respectively) and are likely cross-reactive between species[44,45]. Additionally, the integrin adhesion molecule LFA-1 can interact with mouse ICAM-1[46]. Despite some reported conservation in PC-regulating proteins, several cytokines are not as well conserved between the species. For instance, the BAFF (72% conservation) and BAFFR (54% conservation) interaction is imperfect, as evidenced in engraftment studies showing that human PBMCs into hBAFF knock-in mice led to increased immature B cell development[47]. IL-6 (39% conservation) and IL6R (52%) are also poorly conserved between human and mouse. In a previous study, immunization of human CD34 grafts is greatly improved in hIL-6 knock-in mice[48]. In contrast to such studies, our work was designed to evaluate how human cytokines impact the

retention and durability of mature human PCs; a setting that tests the requirement for human cytokines in establishing and/or maintaining PC longevity. Whereas hIL-6 promoted PC retention/durability in our model, hBAFF also increased BM PC retention. Our findings are in contrast to prior work showing that hBAFF inhibits PC-dependent immune responses in a humanized mouse model[47]. We hypothesize that these differences reflect the expression of hBAFF by the transferred PCs; a setting that may mimic local increases in hBAFF and/or membrane presentation as might occur following immune activation. Our composite findings suggest that additional studies of ex vivo differentiated PCs (in hIL-6 mice or in animals expressing additional human cytokines) will help to identify other key aspects of human PC biology, including surface receptor interactions, niche structure and support cells, and events that modulate competition of PCs into the BM niche. Experiments are in progress to explore the importance of cross-reactivity in additional cytokine (e.g., APRIL; 85% conserved) and receptor (e.g., TACI, BCMA; 51 and 62% conserved) interactions in PC modeling.

Survival of human[6,49] and mouse[50] BM-derived PCs ex vivo is augmented by co-culture with BM stromal cells or treatment with cytokines thought to be present in the BM microenvironment. Additionally, several factors found in BM stromal cell supernatants, including IL-6, promote ex vivo survival of antigen-specific mouse[50] and human[6,51] PCs. Our in vitro observations indicate that IL-6 can promote longevity via anti-apoptotic effects on CD38$^{hi}$CD138$^{lo}$ cells. Consistent with the idea that IL-6 promotes survival prior to terminal PC differentiation, the in vivo effect provided by IL-6 is most pronounced during the first month following the transfer of ex vivo-derived B cells. Similarly, while APRIL, BM stromal cell supernatants, and hypoxia provide a significant immediate survival benefit to the blood-derived tetanus-specific PCs, the decay rate in ex vivo culture is similar across conditions[24]. Together, these results imply that survival benefits conferred by many BM cytokines could occur during PC maturation rather than the following maturation. Finally, it is possible that the persistence of ex vivo-derived PCs in an immune-competent setting could be boosted by treatments that transiently promote hIL-6 secretion, which is a topic of active investigation. Collectively, these data suggest that ex vivo-derived PCs are sufficient to acquire a long-lived phenotype in vivo, but their survival is increased by IL-6-like signals along with APRIL and other possible other BM-derived factors.

Although the full determinants for the acquisition of a long-lived PC phenotype remain to be found, recent evidence suggests that long-lived PCs require alterations in mitochondrial function and glucose uptake[27]. The transcriptional changes we observed in ex vivo-derived PCs cultured for longer time periods are consistent with these prior findings. Although human BM PCs can be either CD19+ or CD19[-6,52], it has been proposed that CD19 loss is required for a long-lived PC phenotype. We observed a decrease in CD19 expression from day 13 to day19 in ex vivo PC culture. However, ex vivo culture-derived, long-lived PCs engrafted in humanized mice retained CD19 expression, albeit at low levels. Thus, loss of CD19 is not a prerequisite for a long-lived human PC phenotype. Finally, a striking observation from our study is that hIL-6 not only promotes long-term PC engraftment, but also functions to increase per-cell antibody secretion rates. The distinct effects of hIL-6 on PC engraftment and protein production both likely contribute to the observed differences in long-term antibody production in the NSG-hIL-6 model and could also contribute to increased secretion of engineered proteins by PC-derived cell therapy.

In summary, our findings help to deconvolute the multiple impacts of hIL-6 on the generation and engraftment of long-lived human PCs. Further, future studies using the ex vivo PC generation and adoptive transfer platform described here, will help to uncover additional fundamental features of long-lived PCs, and in parallel, support the capacity to build and test PC-based cell therapies.

## Methods

### Study design

Deidentified human PBMCs were acquired under informed consent from the Fred Hutch Specimen Processing and Research Cell Bank (protocol #3942). All animal studies were performed according to the Association for Assessment and Accreditation of Laboratory Animal Care (AAALAC) standards and were approved by the SCRI Institutional Animal Care and Use Committee (IACUC).

The aim of this study is to evaluate the requirement of human cytokines for longevity in engrafted ex vivo differentiated human PCs. We differentiated PCs from naive B cells ex vivo and characterized these cells using flow cytometry, RNA sequencing, and transmission electron microscopy. Next, an in vivo experiment was designed to test whether human PCs will engraft in mice in NSG mice with or without synthetic expression of hIL-6 or hBAFF. The in vivo longevity of PCs was assessed by quantifying the abundance of human antibodies, by mathematical modeling and luminescence imaging. We characterized the effect of human IL-6 on PCs by measuring BrdU incorporation, caspase activation, and ELISPOT. Sample sizes in each experiment are clarified in each figure and legend. Each experiment has at least three replicates and ex vivo experiments are designed to have at least three biological replicates from independent donors.

### PC differentiation ex vivo

We isolated B cells from healthy donors' PBMCs (Fred Hutchinson Cancer Research Center) using the EasySep Human B cell isolation kit (Stem Cell Technologies). Isolated B cells were cultured in Iscove's modified Dulbecco's medium (Gibco), supplemented with 2-mercaptoethanol (55$\mu$M) and 10% FBS. Cells were cultured for 7 days (activation) in medium containing 100 ng/mL megaCD40L (Enzo Life Science), 1$\mu$g/ml CpG ODN2006 (Invitrogen), 50 ng/ml IL-2, 50 ng/ml IL-10 (Peprotech), and 10 ng/ml IL-15 (Miltenyi), for 3 days (PB differentiation) in medium containing 50 ng/ml IL-6 (Peprotech) 50 ng/ml IL-2, 50 ng/ml IL-10, and 10 ng/ml IL-15 and 3 days (PC differentiation) in medium containing 50 ng/ml IL-6, 10 ng/ml IL-15, and 15 ng/ml interferon-$\alpha$2B (Sigma-Aldrich)[9].

### Flow cytometry

Flow cytometric analysis was performed on an LSR II flow cytometer (BD Biosciences) and events were analyzed using FlowJo software

(Tree Star). Flow cytometry gating for fluorescent proteins and viability, and immunophenotyping can be found in supplementary Figures.

### CITE-seq sample preparation and sequencing

Following ex vivo differentiation, cells were treated with Ficol to remove the debris. Viable cells were labeled with oligo-conjugated antibodies (Supplementary Table 1) for tracking surface expression and sample identity (HTO, Biolegend, Supplementary Table 1) labeling using the Biolegend Totalseq-B protocol. Next, we sorted viable cells with FACSAria (BD). Sorted cells were loaded into 10X Genomics Chip G (10X Genomics), at 10,000 cells per lane. Next, we prepared libraries using the 10X Genomics Chromium Next GEM Single-Cell v3.1 kit following the 10X user guide (CG000317). Libraries from cell surface tags and transcripts were evaluated by tapestation (Agilent) before sequencing. Finally, libraries were pooled in a ratio of 80% RNA, 20% oligos, and sequenced with NextSeq 1000/2000 kit (Illumina) using the following read length: 28 bp Read1, 10 bp i7 Index, 10 bp i5 Index, and 90 bp Read2.

### Single-cell RNA-seq analysis

Fastq files were processed by CellRanger based on the human reference genome GRCh38. The h5 file is then further analyzed by a python script. Data were demultiplexed by hashsolo[53] and following analysis including normalized, trajectory inference (data clustered using the Louvain-graph-based algorithm and reconstructed the lineage relations by PAGA), and other hierarchical clusterings, dimensional reduction, and cell clustering is analyzed by python package scanpy[54].

### B cell subset bulk RNA sequencing

Following ex vivo differentiation, PCs were isolated using EasySep Human CD138 positive isolation kit and PBs were isolated from the flow-through using positive selection with a CD38 MicroBead Kit (Miltenyi Biotec). Isolated PBs and PCs were stored in Trizol at −80 °C. RNA extraction was performed using an RNeasy mini kit (Qiagen). Following extraction, cDNA synthesis, library preparation, and RNA sequencing was done in Benaroya Research Institute Genomics Core Lab using Clontech's SMARTseq and Illumina's NexteraXT kits.

### RNA-seq analysis

FASTQ files were filtered and adapters were trimmed using Trimmomatic[55]. We then use STAR[56] to map high-quality reads to the human genome reference sequence (version hg38) and associate read counts with each gene. Differential gene expression analysis was performed using DEseq2[57] with a false discovery rate cutoff of 0.05. We also used paired $t$-tests to assess differences between cell types across four different donors. Functional annotation clustering was performed using DAVID (database for annotation, visualization, and integrated discovery)[58,59] online bioinformatic resource with a cluster selection requirement of an enrichment score cutoff of 1.0 and a Benjamini−Hochberg adjusted $p$ value of 0.1.

### CRISPR AAV editing

CRISPR RNAs (crRNAs) and AAV-based homology arms for targeting and gene delivery to the CCR5 locus were identified and validated previously[9]. Briefly, B cells were isolated and activated for 2 days (as described above), and electroporated (Lonza) in batches of 200 million B cells with 4 uM of ribonucleoprotein (RNP) containing sgRNA, trRNA, and Cas9 protein (IDT) in molar ratio = 1:1: 0.83. After electroporation, cells were transferred into an activation medium (1.5 million cells/mL) in the presence of AAV6 encoding *CCR5* repair templates for delivery of firefly luciferase or hBAFF (20% AAV by volume). The medium was changed 18 h following AAV6 administration.

## TEM imaging

Cells were fixed in ½ strength Karnovsky's fixative (2.5% glutaraldehyde, 2% paraformaldehyde in 0.1 M sodium cacodylate buffer, pH 7.3) overnight at 40 °C. After centrifugation, cell pellets were rinsed with 0.1 M cacodylate buffer, treated with 1% osmium tetroxide for 1 h, rinsed with cacodylate buffer, and dehydrated through a graded series of alcohols and propylene oxide. Pellets were then embedded in Eponate12 resin (Ted Pella, Inc) 70 nm ultrathin sections were cut using a Leica EM UC7 ultramicrotome, contrasted with uranyl acetate and lead citrate, and imaged on a Thermo Fisher Talos L120c transmission electron microscope at 120 kV. Digital images were acquired with a Ceta 16 M CMOS 4kx4k digital camera system. For quantification of whether ER exists or not, six lab members were presented with labeled representative images, and 100 mixed unlabeled images from B cells (pooled on day 2 and day 13). Each member scored each unlabeled image based on their assessment of whether the ER content was low (i.e., the image looked like the labeled image of an activated B cell) or high (i.e., the image looked like the labeled image of a PC).

## Animal experiments

NOD.Cg-*Prkdc*$^{scid}$*Il2rg*$^{tm1Wjl}$/SzJ (NSG) mice were purchased from Jackson Laboratory. NOD.CB17-*Prkdc*$^{scid}$*Il2rg*$^{tm1}$/Bcgen (B-NDG) mice and NOD.CB17-*Prkdc*$^{scid}$*Il2rg*$^{tm1}$*Il6*$^{tm1(IL6)}$/Bcgen (humanized IL-6 B-NDG) mice were purchased from Biocytogen. All mice were kept in a designated pathogen-free facility at the Seattle Children's Research Institute (SCRI). In the transplant experiments, 8–12-week-old mice were conditioned with 25 mg/kg Busulfan (Selleckchem) via intraperitoneal injection. Twenty-four hours after conditioning, 10 million ex vivo reprogrammed B cells were delivered to each recipient via retroorbital (Fig. 4) or tail-vein infusion (all other experiments). Mice were bled and were imaged using an IVIS imaging system (Perkin Elmer) for 10 min following subcutaneous injection of luciferin (10 $\mu$L/g) following the indicated schedules. Following sacrifice, the BM cells were collected, phenotyped, and/or cultured ex vivo. Sex was randomized prior to cell transfer; both male and female animals were used in the analysis. Animals were housed at ambient temperature and humidity.

## Analysis of antibody and exogenous protein secretion

ELISA to quantify humanIgG/IgM (Total ELISA kit, Invitrogen) and hBAFF (Human BAFF/BLyS/TNFSF13B ELISA kit, R&D systems) were performed per manufacturer instructions (Supplementary Table 1). Single-cell antibody secretion was measured by ELISPOT (Supplementary Table 1). BM cells (500,000 cells per well) and ex vivo cultured PCs (4000 cells per well) were serially diluted four times, twofold on IgG (H + L) Cross-Adsorbed Goat anti-Human (Invitrogen) precoated MSIPS4W10 plates (Millipore). After 24 h, the plates were washed and treated with Goat Anti-Human IgG-HRP (Southern Biotech). Detection of the secondary antibody was accomplished using peroxidase (AEC Substrate Kit, Vector Laboratories), and spots were read by an ImmunoSpot analyzer (Cellular Technology Limited). Images from each well with spot numbers below 50 were analyzed by Fiji ImageJ (2.1.0). The Macros code for batch analysis can be found in supplementary materials. The analysis detail and rationale are as previously described[35]. All the outputs in the csv format are then concatenated by a python script, and the histogram and kernel density estimate curves were computed by seaborn[60] and matplotlib[61] packages.

## Mathematical model fitting and statistical analysis

We used scipy.optimize to obtain the non-linear least squares regression for the model's unknown parameters. The parameters used in the models are described in Supplementary Table 2. Statistical analysis was performed by scipy.stats, Graphpad or excel. The data were assumed to be Gaussian distributed. Stacked bar graphs were presented as mean ± SD by Graphpad Prism 8 (GraphPad, San Diego, CA). In all the figures showing the dynamics of the antibody level and firefly luciferase, the solid lines represent the mean dynamic and the shadows give 95% confidence intervals.

## Data availability

The Bulk RNA sequencing read data in this study are available on GSE211984, and the single-cell and CITE-seq read data in this study are available on GSE212138. The complete list of all the studies is provided as Source Data. Source data are provided with this paper.

## Code availability

The data were analyzed using published software packages and scripts. A python notebook was used to call routine statistical functions and to organize the data. In addition, the python source code used to perform data analysis is available from GitHub at (https://github.com/Rene2718/Long-term-engraftment-of-engineered-human-plasma-cells-).

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

## Acknowledgements

We would like to thank other members of the James lab for helpful comments and discussion in preparation for this manuscript. We would like to thank the viral core at the Center for Immunity and Immunotherapies at Seattle Children's Research Institute for reagents. We would also like to thank Stephen MacFarlane and the Fred Hutch Cellular Imaging Shared Resource for assistance with microscopy and image analysis. Moreover, we would like to thank Brotman Baty Institute and Benaroya Research Institute for sequencing. Last, we would like to thank

Huiyun Sun for offering help with library QC. This research was supported by the Cellular Imaging Shared Resource (CISR) of the Fred Hutch/University of Washington Cancer Consortium (P30 CA015704). This work was supported in part by the Seattle Children's Research Institute (SCRI) Program for Cell and Gene Therapy (PCGT), the Children's Guild Association Endowed Chair in Pediatric Immunology and the Hansen Investigator in Pediatric Innovation Endowment (to D.J.R.), and the NIH under numbers 5R01CA201135 and R01AI140626 (to R.G.J.).

## Author contributions

R.Y.-H.C., K.L.H., D.J.R., and R.G.J. designed the study. R.Y.-H.C., D.J.R., and R.G.J. wrote the manuscript with advice and help from N.D.C. K.L.H. and E.R.S. established the culture system. R.Y.-H.C. and T.Z. performed phenotyping. K.L.H., C.M.S., A.R.O., T.Z., and S.S. performed in vivo experiments. R.Y.-H.C., K.L.H., C.M.S., and E.R.S. performed the ELISA experiments. I.F.K. performed ddPCR and supervised AAV production. K.L.H., C.M.S., and A.R.O. produced the AAV. N.D.C. performed the TEM experiment and analysis. R.Y.-H.C. wrote the mathematical model with advice from Y.-J.Y. R.Y.-H.C. ran 10X experiment and prepared libraries. R.Y.-H.C. ran the fitting analysis, analyzed 10X scRNA-seq, bulk RNA-seq, and ELISPOT spot analysis. A.R.O. performed IVIS imaging and analysis.

## Competing interests

A patent application has been filed (K.L.H., I.F.K., D.J.R., and R.G.J. "Engraftable cell-based immunotherapy for long-term delivery of therapeutic proteins," US patent application no. US20180282692A1). The remaining authors declare no competing interests.
