## [Peer Review File · Nature Communications]

Ex vivo engineered human plasma cells exhibit robust protein secretion and long-term engraftment in vivoREVIEWER COMMENTS

Reviewer #1 (Remarks to the Author):

This manuscript by Rene Yu-Hong Cheng et. al., presents promising data for the development of ex vivo engineered and differentiated long lived plasma cells in genetic medicine. Previous work from the James lab showed that ex vivo differentiated human PCs could generate serum Ig in immunocompromised mice for up to 21 days after engraftment, and that cells engineered to express BAFF from the CCR5 locus generated higher antibody titers in these animals. In this report, in vitro matured plasma cells are more deeply characterized to show they exhibit transcriptional, metabolic, and ultrastructural features expected in authentic human long lived plasma cells. They also show that provision of human IL-6 in their NSG mouse engraftment model coupled with addition of a BAFF expression cassette into in vitro differentiated LLPCs, greatly enhanced long-term engraftment and Ig secretion in this mouse model. They also show in vitro that IL-6 is likely working by preventing apoptosis of PCs and increasing rates of Ig production.

Questions/comments/concerns

Introduction

1. The authors have omitted reference to the only two previous publications that clearly show durable engraftment of engineered long-lived plasma cells into the BM of mice which should be added to the list at the end of paragraph one. In Moffett et al. Science Immunology 2019 "B cells engineered to express pathogen-specific antibodies protect against infection", engineered LLPCs are identified in the BM of immunocompromised mice 87 days after cell transfer. In Huang et al. Nature Communications 2020 "Elicitation of HIV broadly neutralizing antibodies from engineered B cells", engineered murine LLPCs are sequenced from the bone marrow of immunocompetent mice 250 days after engineered B cell transfer.
2. Given the initial editorial concern over the impact of the advance presented here, perhaps it might be appropriate to introduce potential advantages of in vitro differentiated LLPCs as therapeutics vs. in vivo matured LLPCs, (as in Barzel/Voss demonstrations), thus highlighting the need for a BM engraftment model for potential human LLPC based cell therapy products?

Characterization of in vitro differentiated PCs.

1. It seems like some single cell gene transcription profiling would have been appropriate for the analysis of fully differentiated in vitro matured PCs, where t-SNE plots of CD138+ populations for example, are presented to show heterogeneity of this population after differentiation culture. These various populations of cells could then be compared with those LLPCs that survived in the BM of hIL-6 mice (BAFF engineered or WT) later on. Some deeper characterization of cells that survived in vivo in these animals compared to what was engrafted (beyond cell surface marker t-SNE plots) does seem warranted for this study. On a related note, no mention was made about the diversity of cell morphologies exhibited by ex vivo differentiated PCs. Did all/most live activated or differentiated cells look like the representative images shown in Figure 1G and the supplement? If not, what fraction of cells looked like this?

Ex vivo differentiated PCs home to the BM and stably express hIgG.

1. Radiance signals (or lack thereof) generated in luciferase engineered cell negative control animals after luciferin injection should be at least mentioned with respect to data shown in Figure 2. Perhaps single cell gene transcription profiles of cells that migrate to the chest vs. BM could be profiled at Day 16? Is the durable surviving population in the hindlimb unchanged compared with Day16?
2. (Minor) How were luciferase engineered PCs isolated for engraftment? Or do you mean 10M luciferase engineered cells were engrafted based on your previously determined engineering efficiency (i.e. 10M is 18.6% of the total targeted population engrafted?)

Human IL-6 and BAFF promote engraftment of ex vivo differentiated PCs

1. While IL-6 levels in mice given this cytokine via lentivirus are shown in the supplementary materials, I am not seeing something similar later for the hIL-6 KI mice or a reference showing IL-6

expression in these animals?

2. Why is total flux so low for control animals in Figure 3J vs. the initial experiment in Figure 2B? (Radiance scales have not changed between these data sets?) Dynamics of cell trafficking seems different as well, no radiance surge/IgG spike at D15/16?

3. The data points presented in Figure 4B-D are somewhat confusing given that the experiments were carried out twice with 7 animals each time in control mice, and 8 animals each time in hIL-6 mice? There seems to be inconsistent numbers of data points and I am not sure what data was included here or how it was synthesized for these figures. i.e. the 30 day time point has three data points for hIL6 animals for IgG concentrations vs. maybe 7 overlapping data points for total flux? Is the line the average of data points from all 14/16 animals from both independent experiments? Why wouldn't statistical significance between groups be displayed as the result of appropriate tests as is done for other figures? Statistical significance between groups should be demonstrated in Figure 4F as well.

4. Again, it isn't clear what data is being included in the correlation plot 4G. All control mice here seem to have no IgG serum titer, and the mouse with IgG concentrations at 200ug/ml is missing from C.? I am curious why a high level of human B cell engraftment into the bone marrow of these animals would result in no serum IgG concentration.

hIL6 increases survival of ex vivo differentiated PCs by inhibiting apoptosis of CD138^{lo} PCs

1. How does PC behavior compare in the hIL-6 mouse vs. long term IL-6 culture?

Reviewer #2 (Remarks to the Author):

The paper is primarily a successful engineering effort to produce and manipulate human PCs in vitro and demonstrate their engraftment into preconditioned, immunocompromised mice. This is an impressive achievement showing efficient genetic editing of the cells and in vivo antibody production lasting up to a year. The study is more limited in providing new information on the differentiation of the PCs and their long term survival that could inform future efforts. The study does leverage the model to investigate the mechanisms by which the engraftment, survival and antibody secretion is enhanced by IL-6 and to some degree also BAFF. However, some of the interpretations are complicated by the uncertainty of which cells actually engraft and by the complex interplay between the human cells and the mouse BM environment.

Strong points:

- Establishes that a cytokine-driven in vitro protocol for human PC generation yields populations that can survive long term in NSG mice
- shows that supplementing the NSG mice with human IL-6 improves BM PC engraftment and antibody secretion
- shows that IL-6 promotes survival of plasmablasts and possibly the rate of antibody secretion in PCs

Major criticisms:

While the populations of activated B cells, PBs and PCs generated in vitro are characterized, and there is some phenotyping of the cells that survive in the BM, it is not clear which populations actually engraft the BM after transfer. Since only a single in vitro protocol is used and a mixture of cells is transferred it is difficult to see why the transfer works and how it could be made more efficient in the future.

While the effects of IL-6 are well demonstrated, it is not clear which of the other important factors from the mouse environment are acting on the human cells. The experiments with BAFF are intriguing but also confusing as the BAFF is transduced into the B cells themselves, yet achieving a cell-intrinsic effect. Some clarity on the expected mouse-human cell crosstalk in the BM would help.

The modelling is a nice approach to extract mechanistic information from the data. The description however is often unclear in the main text. For example in the initial introduction of model 1 suggests that the model is capturing dynamics of different cell populations. But from supplemental note 1 it is

not working with different cell populations, there is a single decay component. How were data in Fig S6A produced? Similarly, the description of the multi-component decay in model 2 is hard to understand.

Many of the analyses are either missing statistics or are actually not statistically significant despite being highlighted as important in the text. For example the effects of IL-6 in Fig 4, IgG secretion in Fig 5 and S9.

Minor points:

There is information on the percent of PCR-detected edits. What proportion of cells actually express luciferase and is this proportion the same in all the populations?

The transfer experiments have no controls. It would be good to show some baseline levels of the signals for the luciferase and antibody levels.

Fig 3 E,F would be easier to see on a log scale

Fig S9C (size of PCs) is not referred to in the main text.

Reviewer #3 (Remarks to the Author):

The manuscript "Long-term engraftment of engineered human plasma cells" by Cheng et al. reports that ex vivo differentiated human plasma cells have the potential for long-lived in vivo protein secretion as shown by transplantation in immunodeficient mouse models. The authors demonstrate that engineered human plasma cells exhibit transcriptional and ultrastructural features of long-lived human plasma cells and can migrate to the bone marrow in transplanted immunodeficient NSG mice. In addition, the presence of human cytokines IL6 and BAFF significantly increases the retention of transplanted plasma cells in the bone marrow, with successful engraftment exceeding a year in the recipient mice. The authors also found that human IL6 decreases apoptosis in newly differentiated plasma cells, while promoting per cell antibody production in engineered plasma cells. The authors suggest that their findings support the potential of engineered human plasma cells as a drug-secreting cell therapy for the treatment of disease.

The findings of this manuscript are clearly shown and the data presented in this study are solid. However, the findings described in this manuscript represent an increment in current knowledge and are not considered as a striking advance in the field. Generation and engineering of human long-lived plasma cells have been previously published (Jourdan et al., IL6 supports the generation of human long-lived plasma cells in combination with either APRIL or stromal cell-soluble factors. *Leukemia* 28: 1647-56, 2014; Luo et al., Engineering of α -PD-1 antibody-expressing long-lived plasma cells by CRISPR/Cas9-mediated targeted gene integration. *Cell Death Dis.* 11: 973, 2020). The roles of IL6 and BAFF on the survival or homing of human long-lived plasma cells are well-known and summarized in multiple review articles (Chu and Berek, The establishment of the plasma cell survival niche in the bone marrow. *Immunol. Rev.* 251: 177-88, 2013; Khodadadi et al., The Maintenance of Memory Plasma Cells. *Front. Immunol.* 10: 721, 2019). Successful engraftment in immunodeficient NSG mice have been previously demonstrated for human B cells, T cells, hematopoietic stem cells and a variety of human iPS-derived cells in a number of published studies. Therefore, the findings of this manuscript lack sufficiently striking novelty in concepts, technologies or scientific insights to be published in *Nature Communications*.

REVIEWER COMMENTS

Reviewer #1 (Remarks to the Author):

This manuscript by Rene Yu-Hong Cheng et. al., presents promising data for the development of ex vivo engineered and differentiated long lived plasma cells in genetic medicine. Previous work from the James lab showed that ex vivo differentiated human PCs could generate serum Ig in immunocompromised mice for up to 21 days after engraftment, and that cells engineered to express BAFF from the CCR5 locus generated higher antibody titers in these animals. In this report, in vitro matured plasma cells are more deeply characterized to show they exhibit transcriptional, metabolic, and ultrastructural features expected in authentic human long lived plasma cells. They also show that provision of human IL-6 in their NSG mouse engraftment model coupled with addition of a BAFF expression cassette into in vitro differentiated LLPCs, greatly enhanced long-term engraftment and Ig secretion in this mouse model. They also show in vitro that IL-6 is likely working by preventing apoptosis of PCs and increasing rates of Ig production.

Questions/comments/concerns

Introduction

1. The authors have omitted reference to the only two previous publications that clearly show durable engraftment of engineered long-lived plasma cells into the BM of mice which should be added to the list at the end of paragraph one. In Moffett et al. Science Immunology 2019 “B cells engineered to express pathogen-specific antibodies protect against infection”, engineered LLPCs are identified in the BM of immunocompromised mice 87 days after cell transfer. In Huang et al. Nature Communications 2020 “Elicitation of HIV broadly neutralizing antibodies from engineered B cells”, engineered murine LLPCs are sequenced from the bone marrow of immunocompetent mice 250 days after engineered B cell transfer.

Thank you for pointing this out - we agree and have added the missing references and incorporated a better description of these important findings to the introduction (see lines 42-47).

2. Given the initial editorial concern over the impact of the advance presented here, perhaps it might be appropriate to introduce potential advantages of in vitro differentiated LLPCs as therapeutics vs. in vivo matured LLPCs, (as in Barzel/Voss demonstrations), thus highlighting the need for a BM engraftment model for potential human LLPC based cell therapy products?

Thank you - we have attempted to contextualize both approaches and have added a sentence to the introduction that highlights the therapeutic reasons for developing a cell therapy based on engineered ex vivo differentiated plasma cells. We see the primary application of ex vivo generated B cell therapeutics to be delivery of non-antiviral biologics requiring stable, long term, and unmutated delivery. The obvious advantage of the in vivo maturation approach is to produce a “living” antibody-based therapeutic that can either be the source for vaccine priming, or mature and adapt upon viral challenge. We envision these two approaches, among many

other possible applications for engineered B cells, to be complementary, and to likely target different disease classes. See lines 49-54 and 64-71.

Characterization of *in vitro* differentiated PCs.

1. It seems like some single cell gene transcription profiling would have been appropriate for the analysis of fully differentiated *in vitro* matured PCs, where t-SNE plots of CD138⁺ populations for example, are presented to show heterogeneity of this population after differentiation culture. These various populations of cells could then be compared with those LLPCs that survived in the BM of hIL-6 mice (BAFF engineered or WT) later on. Some deeper characterization of cells that survived *in vivo* in these animals compared to what was engrafted (beyond cell surface marker t-SNE plots) does seem warranted for this study.

Thank you for this insightful comment; we agree that the cell surface phenotyping data provides an incomplete picture of the heterogeneity in the differentiated B cell cultures. To address this, we used scRNA sequencing combined with surface labeling (CITEseq) to characterize the cell product at two timepoints; this data is presented in new Figure 1 and new Supplementary Figure 1. Briefly, the new data indicate that after differentiation, several cell types exist *in vitro*. The populations that express PC transcriptional phenotypes also express CD138 and/or CD38. The primary discriminator between PCs (CD38^{hi}CD138^{hi} or CD38^{hi}CD138^{lo}) is the specific antibody isotype and/or light chain. However, we did notice interesting differences between IgG and IgM/A populations, most notably that there are fewer CD138^{hi} cells among IgM/A PCs. The gene expression comparison between the surface phenotypes (CD38^{hi}CD138^{hi} versus CD38^{hi}CD138^{lo}) is presented for selected genes in Figure 1H and the population Supplementary Figure 1.

This analysis enabled us to make additional observations implying that *ex vivo* differentiated PCs are similar in many ways to *in vivo* derived PCs. We carefully elaborate upon that data in the text (lines 80-133).

Because of the low numbers of PCs that engraft in our mouse studies, it isn't feasible or straightforward to characterize cells using scRNA sequencing; fewer than .02% of bone marrow cells are human PCs. However, we have characterized the durability of IgM, IgG and IgA plasma cells using flow cytometry, ELISA and ELISPOT using cells isolated at the time of sacrifice. To further query the *in vivo* PCs, upon sacrifice of the IL6 experiment presented in the updated Figure 5, we ran intracellular flow and backgated on the IgG and IgM plasma cells to determine their CD138 phenotype. We found that the CD138^{hi} cells were predominantly IgG⁺ (reviewer figure below). Furthermore, in the subset study described below, IgG⁺ PCs make up the vast large majority of engrafted CD138⁺ cells. This implies that the IgG⁺ LLPCs present post-transfer are most likely derived from *in vitro* generated CD138^{hi}, IgG⁺ cells.

Fig. R1. Transfer of *ex vivo* derived PCs into B-NDG (control; 2 independent experiments, n=7) and hIL6-B-NDG knock-in (hIL6; 2 independent experiments, n=8) mice. Mice were sacrificed after a month and BM cell enriched. Human cells are gated by hCD45+ and mCD45-, the PCs were gated by hCD38+. Bar plot present percentage of antibody isotypes in human CD138lo PCs vs CD138hi PCs in BM from sacrificed mice.

To address reviewer 2, we isolated the CD138- and CD138+ cells using bead-based enrichment after differentiation and infused both populations into mice. Similar to our contention above, we found that the CD138+ PCs engraft similar to unenriched PCs, and that CD138- PCs engrafted poorly (Fig. 3E-3G).

Fig. R2 (new Fig 3E-G). Day 13 *ex vivo* culture cells were enriched by anti-CD138 beads, flow cytometry of cell subsets after enrichment with CD138 antibody is shown in (E). And followed infused isolated cells to mice. (F) Representative images and (G) quantification from *in vivo* subset engraftment tracking engineered B cells after a month. Cells were edited to express firefly luciferase and were imaged by IVIS.

Taken together, the combined new data strongly support the conclusion that the engrafted LLPC populations identified *in vivo* are primarily derived from CD138+CD38+ PCs present within the *in vitro* differentiated B cell pool. See lines 181-193.

On a related note, no mention was made about the diversity of cell morphologies exhibited by ex vivo differentiated PCs. Did all/most live activated or differentiated cells look like the representative images shown in Figure 1G and the supplement? If not, what fraction of cells looked like this?

To address this critique, we quantified the cells by morphology. The primary morphological difference between the PCs and activated B cells was the quantity of ER/golgi in the cells. We collected 50 EM images from cells taken from the activated B cell cultures (day 2) and 50 images from the PC cultures (day 13). Six lab members independently scored each unlabeled image based on their assessment of whether the ER content was low (ie the image looked like the labeled image of an activated B cell) or high (ie the image looked like the labeled image of a PC). There was high concordance between individual scores of the unlabeled images. Approximately 50% of cells within the day13 cultures exhibited high-ER content by this metric, versus zero in the day 2 cultures. This proportion of ER-high cells at Day 13 closely parallel our quantification of PCs via both flow phenotyping and via scRNA transcriptional signature (the quantification is presented in Fig. 2D).

Fig. R3. Quantification of Transmission electron microscopy of day 2 and day 13 B cells from three-stage culture. Rough endoplasmic reticulum (RER) in ex vivo differentiated cultures. > 40 images of day 2 and day 13 cells were scored blindly for having prominent RER or not.

Ex vivo differentiated PCs home to the BM and stably express hlgG.

1. Radiance signals (or lack thereof) generated in luciferase engineered cell negative control animals after luciferin injection should be at least mentioned with respect to data shown in Figure 2.

We apologize that we didn't include negative controls in this experiment. To address this deficit, we collected radiance signals in historically similar experiments using PBS injections (no cell control). In brief, the baseline total flux (negative control) is consistently measured at approximately 6×10^5 (p/s). For reference, we typically observe 4×10^6 - 2×10^7 flux values in

experiments that have cells (see Fig.3C). Finally, in the new engraftment experiment (Fig. 3G), we included the PBS injected negative control.

Fig. R4. Bar and scatter plot of total flux of negative control (PBS) from each of mouse imaging experiment historically.

Perhaps single cell gene transcription profiles of cells that migrate to the chest vs. BM could be profiled at Day 16? Is the durable surviving population in the hindlimb unchanged compared with Day16?

Thank you for this comment; we agree that a comparison of the sternum and hindlimb cell populations might provide interesting information. However, because fewer than .02% of bone marrow cells are human PCs, live cell scRNA sequencing is not feasible. Importantly, the percentage of total flux in hindlimb vs. chest remain unchanged after day16.; and while both decline slightly after day16 (see Figure 3C), the relative proportions in each area remained constant over time (see below).

Fig. R5. *In vivo* engraftment experiment tracking engineered B cells from Day 0 to day 51 using firefly luciferase imaging by IVIS (n = 10, 2 independent donors.) Percentage of total flux in hindlimb and chest region over time.

2. (Minor) How were luciferase engineered PCs isolated for engraftment? Or do you mean 10M luciferase engineered cells were engrafted based on your previously determined engineering efficiency (i.e. 10M is 18.6% of the total targeted population engrafted?)

We calculated the HDR rates using ddPCR from gDNA taken from the cell product just prior to infusion. To clarify, of the 10M cells infused, with an 18.6% HDR allele frequency, slightly more than ~1.8M were likely luciferase positive. We have edited the main text for clarification. (See lines 166-167).

Human IL-6 and BAFF promote engraftment of ex vivo differentiated PCs

1. While IL-6 levels in mice given this cytokine via lentivirus are shown in the supplementary materials, I am not seeing something similar later for the hIL-6 KI mice or a reference showing IL-6 expression in these animals?

Thank you for this question. Since this mouse model is commercially available, we didn't initially provide the data. However, prior to working with the model, we sacrificed NSG and hIL6-NSG mice, isolated BM cells and treated them with golgi stop for an hour prior to staining with antibodies directed against mouse and human IL6. As you can observe below, human IL6 is only detectable within the hIL6-NSG mice (below left). Additionally, we elicited IL6 secretion using *in vivo* administration of LPS and quantified serum IL6. Again, human IL6 is only detected in hIL6-NSG mice. It should be noted that unlike the lentiviral model, IL6 is only detectable in serum during brief periods following stimulation (below right). Thus, we consider the hIL6-NSG model to be more physiologically relevant than the lentivirus model.

Fig. R6. (Left) Flow plot of isolated BM cells from hIL6 mouse and control mouse using antibodies directed against mouse and human IL6. Note that detection of human and murine IL6 is exclusive to hIL6 and B6 mice respectively. (Right) hIL6 and control (B6) mice were injected with the indicated quantities of LPS or with PBS. After 2 hours, we quantified human IL6 from serum.

2. Why is total flux so low for control animals in Figure 3J vs. the initial experiment in Figure 2B? (Radiance scales have not changed between these data sets?) Dynamics of cell trafficking seems different as well, no radiance surge/IgG spike at D15/16?

Thank you for pointing this out. In the original presentation, we made a mistake and plotted the figures using 2 different radiance scales. However, in addition to the scaling issue, there is some experiment-experiment variability caused by the degree of gene editing, engraftability of the donor and differences in timing of the luciferase imaging. To improve clarity, we revised our Figures using the same units (new Fig 2F and Figure 3C, 3G). In revised Figure 2F, the flux levels are slightly lower than in 3B-C (~50% total flux). Data that predated the more physiological hIL6 model has been moved to supplement (new Figure S6E-G) and revised Figure 4B (new Figure 5B) present data collected from 2 experiments on the same plot. We normalized the flux based on aggregate values (min-max normalization), quantified across the entire experiment, and then plotted the data together.

The difference in dynamics is an interesting question. We have observed donor related variation that likely leads to variable proportions of short-lived PCs generated in vitro. These differences may reflect differences in input B cell populations including the proportion of memory, transitional and naive cells. This is an area of active investigation by our lab.

3. The data points presented in Figure 4B-D are somewhat confusing given that the experiments were carried out twice with 7 animals each time in control mice, and 8 animals each time in hIL-6 mice? There seems to be inconsistent numbers of data points and I am not sure what data was included here or how it was synthesized for these figures. i.e. the 30 day time point has three data points for hIL6 animals for IgG concentrations vs. maybe 7 overlapping data points for total flux? Is the line the average of data points from all 14/16 animals from both independent experiments? Why wouldn't statistical significance between groups be displayed as the result of appropriate tests as is done for other figures? Statistical significance between groups should be demonstrated in Figure 4F as well.

We apologize for the confusing presentation of these data. In the original presentation, Figure 4C (currently Figure 5C) had one dot outside of the 95% Confidence interval. When we initially made the figure, we did not include this data point because it was outside of the range presented in the figure. In the updated figure (Figure 5C), the scale has been amended and the data point is now included.

In panels B-D and G, the data was presented from 8 hIL6 mice (combined from 2 experiments) and 7 control mice (combined from 2 experiments). The individual spots are sometimes obscured by other spots. The normalization in panel B (described above) is now explained in the legend. There is no normalization used for panels C and D; the lines represent averages and the shaded areas represent the range between the 95% confidence intervals. By some definitions, if 95% confidence intervals do not overlap, datasets can be considered significantly different. To further address significance, we calculated area under the curve values for

individual mice for flux, IgG and IgM and performed unpaired t-tests. The p-values associated with those tests are now reported in the figure.

For each of the subpopulations described in Figure 5F, we calculated p-values using unpaired t-tests. As shown below, we observed that some subsets (ASC, IgG, IgM and CD138lo) are significantly more abundant in IL6 animals relative to controls (alpha = 0.1). We added a discussion of the statistical tests in the results (lines 279-281). Note that because of the nature of the bone marrow harvest, we do expect higher variance in the cell subset enumeration than in other metrics (e.g. total IgG or flux).

	ASC	IgG	IgM	IgA	CD138lo	CD138hi
p-value	0.064	0.098	0.023	0.13	0.044	0.11

4. Again, it isn't clear what data is being included in the correlation plot 4G. All control mice here seem to have no IgG serum titer, and the mouse with IgG concentrations at 200ug/ml is missing from C.? I am curious why a high level of human B cell engraftment into the bone marrow of these animals would result in no serum IgG concentration.

There are several possible explanations for why the level of human B cell engraftment might not correlate with IgG abundance. First, the cell numbers required to produce the quantities of IgG that we have observed in Figure 5 are very small (~.02% of human cells). Because of the nature of the cell harvests (1 femur, lots of handling), the small numbers likely lead to high variance simply due to handling alone. Therefore, we think it makes sense that there is only a correlation between cell numbers and IgG when there are high quantities of IgG and therefore high quantities of cells. Despite the potential issues with handling, we still observed statistically significant differences between PC cellularity in NSG and NSG-hIL6 mice (described in lines 310-311 of the text). Second, it is possible that there is additional biology at play in this system; for example, hIL6 appears to elicit increased IgG production on a per ASC basis (Figure 6). Separating the role of IL6 in the regulation of protein production versus enumeration/engraftment is challenging, especially in the context of the sample handling variance described above. Although these secretion differences contribute to the findings observed here, we chose not to speculate due to the inherent variability in the assay.

hIL6 increases survival of ex vivo differentiated PCs by inhibiting apoptosis of CD138lo PCs
1. How does PC behavior compare in the hIL-6 mouse vs. long term IL-6 culture?

We thank the reviewer for this comment. The increase in engrafted cell numbers shortly after transfer (current Figure 4) is consistent with a role for hIL6 in extending the survival of PBs; presumably via supporting differentiation *in vivo* to provide a larger pool of LLPCs. To try and partially address this question, we assessed the *in vivo* dynamic in the hIL6 model (new Fig. 5B and reviewer figure below). Consistent to our observations in the LV-IL6 model, the most

striking difference occurred in the ratio of ffluc (hIL6 vs control) within 72 hr of cell transfer (figure shown below). These findings are consistent with our observations in long term IL-6 cultures.

Fig. R7. Transfer of *ex vivo* derived PCs into B-NDG (control; 2 independent experiments, n=7) and hIL6-B-NDG knock-in (hIL6; 2 independent experiments, n=8) mice. Firefly luciferase was quantified by IVIS imaging. Dynamic of ratio of mean of hIL6 mice total flux and mean of control mice total flux.

Reviewer #2 (Remarks to the Author):

The paper is primarily a successful engineering effort to produce and manipulate human PCs in vitro and demonstrate their engraftment into preconditioned, immunocompromised mice. This is an impressive achievement showing efficient genetic editing of the cells and in vivo antibody production lasting up to a year. The study is more limited in providing new information on the differentiation of the PCs and their long term survival that could inform future efforts. The study does leverage the model to investigate the mechanisms by which the engraftment, survival and antibody secretion is enhanced by IL-6 and to some degree also BAFF. However, some of the interpretations are complicated by the uncertainty of which cells actually engraft and by the complex interplay between the human cells and the mouse BM environment.

Strong points:

- Establishes that a cytokine-driven in vitro protocol for human PC generation yields populations that can survive long term in NSG mice
- shows that supplementing the NSG mice with human IL-6 improves BM PC engraftment and antibody secretion
- shows that IL-6 promotes survival of plasmablasts and possibly the rate of antibody secretion in PCs

Major criticisms:

While the populations of activated B cells, PBs and PCs generated in vitro are characterized, and there is some phenotyping of the cells that survive in the BM, it is not clear which populations actually engraft the BM after transfer. Since only a single in vitro protocol is used and a mixture of cells is transferred it is difficult to see why the transfer works and how it could be made more efficient in the future.

We thank the reviewer for this important comment. To address this question, we isolated the CD138⁻ vs. CD138⁺ cell populations (using bead-based enrichment after differentiation) and transferred the alternative populations into mice. As predicted in the contention above, we found that the CD138⁺ PCs engrafted rates comparable to that of unenriched PCs. In contrast, CD138⁻ IgG⁺ PCs engrafted less efficiently (Fig. 3E-3G). Taken together, the new data we present implies that the IgG⁺ LLPC populations within the differentiated B cells are likely CD138⁺CD38⁺ PCs. See lines 181-193.

Based on this new engraftment data, we predict that the protocol could be improved by identifying cytokine cocktails that more effectively promote differentiation into CD138⁺CD38⁺ PCs. This is an active area of investigation by our labs.

Fig. R8 (new Fig 3E-G). Day 13 ex vivo culture cells were enriched by anti-CD138 beads, flow cytometry of cell subsets after enrichment with CD138 antibody is shown in (E). And followed infused isolated cells to mice. (F) Representative images and (G) quantification from *in vivo* subset engraftment tracking engineered B cells after a month. Cells were edited to express firefly luciferase and were imaged by IVIS.

While the effects of IL-6 are well demonstrated, it is not clear which of the other important factors from the mouse environment are acting on the human cells. The experiments with BAFF are intriguing but also confusing as the BAFF is transduced into the B cells themselves, yet achieving a cell-intrinsic effect. Some clarity on the expected mouse-human cell crosstalk in the BM would help.

We thank the reviewer for this comment. We agree that the BAFF experiments are somewhat difficult to interpret because the cell intrinsic effect seems more pronounced than the cell extrinsic effect. This is likely due to the mechanism of BAFF-BAFFR/TACI interactions. Functional BAFF requires oligomerization that is challenging to elicit in solution using commercially available recombinant BAFF reagents (monomeric BAFF) compared with the secretion of mature BAFF. Cellular expression of BAFF may promote BAFF-R or TACI cross-linking via a different mechanism. While we agree that these are important questions, we argue that a detailed mechanistic investigation of the impact of secreted BAFF is beyond the scope of this manuscript.

To directly address the second part of this comment, we added a small section to the discussion (lines 376-400) describing our expectations regarding mouse-human crosstalk for key signals that regulate PC function (APRIL, CXCL12, integrins).

The modeling is a nice approach to extract mechanistic information from the data. The description however is often unclear in the main text. For example in the initial introduction of model 1 suggests that the model is capturing dynamics of different cell populations. But from supplemental note 1 it is not working with different cell populations, there is a single decay component. How were data in Fig S6A produced? Similarly, the description of the multi-component decay in model 2 is hard to understand.

We thank the reviewer for pointing out this lack of clarity. We have now included a more detailed explanation of the mathematical models in the main text. Regarding the first model, we focused on the long-lived PC population, because the short-lived PC is not stably retained.

In Fig. S6A, we used equation 1 to fit the long-lived PC and short-lived PC separately. To do this, we fit the long-lived PC population with the IgG dynamic from the third month to the end of the study (“Long-lived ASC fitting; shaded area) and fit the short-lived PC by calculating the dynamic of IgG after subtracting the data from the long-lived PC’s fitting. The result of the second fit is represented in the shaded area in the middle panel (“Short-lived ASC fitting”). Upon publication, the jupyter notebook used for the calculations will be posted on github; in the meantime, we can make this available to the reviewer upon request. We added more description of these models to lines 229-240 in the text.

The second model involved variables for the cell number decay and the decay rate. Since the decay rate is also a function of time, we may have presented the model in a confusing way. To address this comment, we clarified how we generated the model in the main text. In a nutshell, the PC decay rate immediately after day 13 culture is much more rapid than several days after day 13 (compare slope of red and blue lines between day 0-10 to days 10-20 post day 13, Fig. 6B). Using the mathematical model that took into account changing decay rates over time, we found that only a subset of newly generated PCs are retained after day 13 (and likely in vivo). We are currently actively investigating the determinants that facilitate survival of this small subset of cells; the data produced here imply that IL6 is one contributing factor. We added more description of this model in lines 294-302 in the text.

Many of the analyses are either missing statistics or are actually not statistically significant despite being highlighted as important in the text. For example the effects of IL-6 in Fig 4, IgG secretion in Fig 5 and S9.

Thank you for this comment. Please see the response to Reviewer 1, minor comment 3. We have tried to address this concern.

Additionally, in new Figure 6D (old figure 5D), we’ve added statistical tests. For the secretion data originally presented in Fig. S9, we’ve moved the histogram of elispot size to the main figure (S9A → new Fig. 6I). The way we analyzed the difference was using KULLBACK-LEIBLER divergence, but the calculated divergence did not assess significance. Therefore, we took the mean from each distribution and ran t-tests for IgG spot size using the mean values as in Fig. 6J and Fig. 6L shown. Note that the distributions in spot size (example, Fig. 6I) exhibited significant differences in spread across experiments, but in all cases the hIL6 condition exhibited far more frequent large spots. A comparison of the means is unlikely to be the best statistical approach.

Minor points:

There is information on the percent of PCR-detected edits. What proportion of cells actually express luciferase and is this proportion the same in all the populations?

Previously, we have tested GFP-T2A-BAFF editing in B cells CCR5 region with the same editing protocol (Hung et al). When the HDR editing rate by ddPCR was about 25-30%, we observed GFP expression in ~35% of the edited cell population. We have reproducibly observed slightly higher HDR rates in cells with a PC phenotype (~5% higher; also described in Hung et al). Here,

using ddPCR we show that the luciferase HDR allele frequency is ~20%. Based on a 20% allele frequency, we expect that slightly greater than 20% of cells will express luciferase (the majority of edited cells will be heterozygous for HDR). However, since we did not co-deliver a fluorescent reporter and our firefly luciferase antibodies exhibit poor signal:noise in flow cytometry, we cannot give a precise quantification of cells that express luciferase.

The transfer experiments have no controls. It would be good to show some baseline levels of the signals for the luciferase and antibody levels.

We apologize that we don't have negative control from this experiment. We have radiance signals collected historically with PBS (no cell control) after luciferin injection from several experiments previously shown below. We have observed that baseline total flux (negative control) consistently and tightly around 6×10^5 . Also, in the new engraftment experiment (Fig. 3F-G), we included this negative control and quantified the flux (line on Fig. 3G).

Fig. R9. Bar and scatter plot of total flux of negative control (PBS) from each of mouse imaging experiment historically.

Fig 3 E,F would be easier to see on a log scale
Thank you - this has been updated.

Fig S9C (size of PCs) is not referred to in the main text.
Thank you for pointing this out. We removed the data since it addresses a minor point.

Reviewer #3 (Remarks to the Author):

The manuscript "Long-term engraftment of engineered human plasma cells" by Cheng et al. reports that ex vivo differentiated human plasma cells have the potential for long-lived in vivo protein secretion as shown by transplantation in immunodeficient mouse models. The authors demonstrate that engineered human plasma cells exhibit transcriptional and ultrastructural features of long-lived human plasma cells and can migrate to the bone marrow in transplanted immunodeficient NSG mice. In addition, the presence of human cytokines IL6 and BAFF significantly increases the retention of transplanted plasma cells in the bone marrow, with successful engraftment exceeding a year in the recipient mice. The authors also found that human IL6 decreases apoptosis in newly differentiated plasma cells, while promoting per cell antibody production in engineered plasma cells. The authors suggest that their findings support the potential of engineered human plasma cells as a drug-secreting cell therapy for the treatment of disease.

The findings of this manuscript are clearly shown and the data presented in this study are solid. However, the findings described in this manuscript represent an increment in current knowledge and are not considered as a striking advance in the field. Generation and engineering of human long-lived plasma cells have been previously published (Jourdan et al., IL6 supports the generation of human long-lived plasma cells in combination with either APRIL or stromal cell-soluble factors. *Leukemia* 28: 1647-56, 2014; Luo et al., Engineering of α -PD-1 antibody-expressing long-lived plasma cells by CRISPR/Cas9-mediated targeted gene integration. *Cell Death Dis.* 11: 973, 2020). The roles of IL6 and BAFF on the survival or homing of human long-lived plasma cells are well-known and summarized in multiple review articles (Chu and Berek, The establishment of the plasma cell survival niche in the bone marrow. *Immunol. Rev.* 251: 177-88, 2013; Khodadadi et al., The Maintenance of Memory Plasma Cells. *Front. Immunol.* 10: 721, 2019). Successful engraftment in immunodeficient NSG mice have been previously demonstrated for human B cells, T cells, hematopoietic stem cells and a variety of human iPS-derived cells in a number of published studies. Therefore, the findings of this manuscript lack sufficiently striking novelty in concepts, technologies or scientific insights to be published in *Nature Communications*.

We thank the reviewer for the positive comments regarding the nature of the data presented in this study. However, we disagree about the assessment of their novelty. In our opinion, the conclusions described in the above review oversimplify the referenced data. In vivo modeling of human plasma cells is, in fact, a new field. The papers referenced by reviewer 1 (Huang, et al, Moffett et al, Nahmad et al), the Luo paper and our initial manuscript (Hung et al) are the only experiments published to date that demonstrate in vivo engraftment of engineered PCs (all since 2018). Of these, only the Luo and Hung papers showed human PC engraftment and the longest duration described for human PCs was ~150 days.

Further, none of the work cited by the reviewer has assessed the role for IL6, BAFF or April on in vivo human PC biology. Indeed, it remains unclear if, and/or how, these factors impact in vivo longevity, or homing of human PCs. Prior to our current study, there have not been reliable

animal models to study human PCs. Further, the studies in the review articles mentioned by the reviewer are correlative, from mice and/or from in vitro experiments. All relevant in vivo studies regarding IL6 and BAFF reported to date are cited in our paper. This previous work has primarily focused on B cell development from hematopoietic stem cell (HSC) grafts in humanized mice. This latter topic is an entirely different scientific question than the questions addressed in our work- e.g., assessment of the homing and engraftment features of *ex vivo* engineered plasma cells.

Finally, this paper is not attempting to address NSG modeling of mature B cells, iPSCs, T cells or hematopoietic stem cells; thus, we are not sure how this comment is relevant.

REVIEWERS' COMMENTS

Reviewer #1 (Remarks to the Author):

The revised manuscript has been greatly improved in quality. The authors were responsive to original reviewer concerns. They have added new data sets that buttress the original findings and offer some new insights. In particular, the addition of CITE-seq data shows high-resolution details about the ex vivo differentiated human plasma cell cultures. This is especially interesting if such cultures are under consideration for clinical development. The article is suitable for publication in Nature Communications in my opinion, because a well characterized mouse model for the long-term engraftment of engineered human plasma cells will be broadly useful.

Reviewer #2 (Remarks to the Author):

The revised manuscript is much more interesting and provides more insight. The new data on the transcriptomics of the B cell populations and plasma cells and the additional data on the in vivo engrafted populations address my previous comments. The improved description of the modelling and edits in the discussion help clarity. I recommend the paper for publication. My only minor comment is that there is occasional misnumbering of the figures in the text.

Reviewer #3 (Remarks to the Author):

The authors have adequately addressed my previous concerns and critiques in the revised manuscript.

REVIEWERS' COMMENTS

Reviewer #1 (Remarks to the Author):

The revised manuscript has been greatly improved in quality. The authors were responsive to original reviewer concerns. They have added new data sets that buttress the original findings and offer some new insights. In particular, the addition of CITE-seq data shows high-resolution details about the ex vivo differentiated human plasma cell cultures. This is especially interesting if such cultures are under consideration for clinical development. The article is suitable for publication in Nature Communications in my opinion, because a well characterized mouse model for the long-term engraftment of engineered human plasma cells will be broadly useful.

Thank you.

Reviewer #2 (Remarks to the Author):

The revised manuscript is much more interesting and provides more insight. The new data on the transcriptomics of the B cell populations and plasma cells and the additional data on the in vivo engrafted populations address my previous comments. The improved description of the modelling and edits in the discussion help clarity. I recommend the paper for publication. My only minor comment is that there is occasional misnumbering of the figures in the text.

Thank you. We've corrected the misnumbering.

Reviewer #3 (Remarks to the Author):

The authors have adequately addressed my previous concerns and critiques in the revised manuscript.

Thank you.